# Interaction between games give rise to the evolution of moral norms of cooperation

**Mohammad Salahshour** [1,2,3,4] *

**1** Max Planck Institute for Mathematics in the Sciences, Leipzig, Germany, **2** Max Planck Institute of Animal Behaviour, Radolfzell, Germany, **3** Centre for the Advanced Study of Collective Behaviour, University of Konstanz, Konstanz, Germany, **4** Department of Biology, University of Konstanz, Konstanz, Germany

* salahshour.mohammad@gmail.com

**Data Availability Statement:** All relevant data are within the manuscript and its Supporting information files.

**Funding:** The author acknowledges funding from Alexander von Humboldt Foundation in the framework of the Sofja Kovalevskaja Award

## Abstract

In many biological populations, such as human groups, individuals face a complex strategic setting, where they need to make strategic decisions over a diverse set of issues and their behavior in one strategic context can affect their decisions in another. This raises the question of how the interaction between different strategic contexts affects individuals' strategic choices and social norms? To address this question, I introduce a framework where individuals play two games with different structures and decide upon their strategy in a second game based on their knowledge of their opponent's strategy in the first game. I consider both multistage games, where the same opponents play the two games consecutively, and reputation-based model, where individuals play their two games with different opponents but receive information about their opponent's strategy. By considering a case where the first game is a social dilemma, I show that when the second game is a coordination or anti-coordination game, the Nash equilibrium of the coupled game can be decomposed into two classes, a defective equilibrium which is composed of two simple equilibrium of the two games, and a cooperative equilibrium, in which coupling between the two games emerge and sustain cooperation in the social dilemma. For the existence of the cooperative equilibrium, the cost of cooperation should be smaller than a value determined by the structure of the second game. Investigation of the evolutionary dynamics shows that a cooperative fixed point exists when the second game belongs to coordination or anti-coordination class in a mixed population. However, the basin of attraction of the cooperative fixed point is much smaller for the coordination class, and this fixed point disappears in a structured population. When the second game belongs to the anti-coordination class, the system possesses a spontaneous symmetry-breaking phase transition above which the symmetry between cooperation and defection breaks. A set of cooperation supporting moral norms emerges according to which cooperation stands out as a valuable trait. Notably, the moral system also brings a more efficient allocation of resources in the second game. This observation suggests a moral system has two different roles: Promotion of cooperation, which is against individuals' self-interest but beneficial for the population, and promotion of organization and order, which is at both the population's and the individual's self-interest. Interestingly, the latter acts like a Trojan horse: Once established out of individuals' self-interest, it brings the former with itself. Importantly, the fact that the evolution of moral norms depends only on the

endowed by the German Federal Ministry of Education and Research and the Deutsche Forschungsgemeinschaft (German Research Foundation) under Germany's Excellence Strategy EXC 2117-422037984 for funding during parts of this research. The funders had no role in study design, data collection and analysis, decision to publish, or preparation of the manuscript.

**Competing interests:** The authors have declared that no competing interests exist.

cost of cooperation and is independent of the benefit of cooperation implies that moral norms can be harmful and incur a pure collective cost, yet they are just as effective in promoting order and organization. Finally, the model predicts that recognition noise can have a surprisingly positive effect on the evolution of moral norms and facilitates cooperation in the Snow Drift game in structured populations.

## Author summary

How do moral norms spontaneously evolve in the presence of selfish incentives? An answer to this question is provided by the observation that moral systems have two distinct functions: Besides encouraging self-sacrificing cooperation, they also bring organization and order into the societies. In contrast to the former, which is costly for the individuals but beneficial for the group, the latter is beneficial for both the group and the individuals. A simple evolutionary model suggests this latter aspect is what makes a moral system evolve based on the individuals' self-interest. However, a moral system behaves like a Trojan horse: Once established out of the individuals' self-interest to promote order and organization, it also brings self-sacrificing cooperation.

## Introduction

Although beneficial for the group, cooperation is costly for the individuals, and thus, constitutes a social dilemma: Following their self-interest, individuals should refrain from cooperation. This leaves everybody worse off than if otherwise, all had cooperated [1–4]. Indirect reciprocity is suggested as a way out of this dilemma [3, 5–8], which can also bring insights into the evolution of morality [6, 7]. Most models of indirect reciprocity consider a simple strategic setting where individuals face a social dilemma, commonly modeled as a Prisoner's Dilemma. Individuals decide upon their strategy in the social dilemma based on their opponent's reputation. In turn, reputation is built based on the strategy of the individuals in the same social dilemma. This self-referential structure can give rise to some problems. The core of these problems relies on how to define good and bad. In the simplest indirect reciprocity model, only first-order moral assessment rules are allowed: For instance, an individual's reputation is increased by cooperation, and it is decreased by defection [9, 10]. This leads to a situation where defection with someone with a bad reputation leads to a bad reputation. From a moral perspective, this does not make sense. Besides, this can lead to the instability of the dynamic [6]. To solve these problems, it is possible to consider second-order moral assessment rules [11–15]. In prescribing an individual's reputation, besides its action, second-order rules also take the reputation of its opponent into account. However, this way opens the door to third-order and higher-order moral assessment rules [16–19], which require having information about the actions of the individuals further and further into the past [6, 19, 20]. This leads to a rapid increase in the number and complexity of moral assessment rules by going to higher-order rules, even when, as it is commonly assumed, moral assessment is reduced to a binary world of good and bad [6, 17–21].

In contrast to the premise of most models of indirect reciprocity, strategic interactions in many biological contexts are not simple. Individuals in populations often need to solve different strategic problems simultaneously. For instance, they may need to decide whether incur a cost for others to benefit, resolve conflicts to avoid mutual losses [22], coordinate their actions

with others [22–25], for instance by deciding at what time or place engage in an activity, or they may need to choose partners and mates [26]. Importantly, the strategic structures of these diverse activities are generally distinct. While some entail a pure cost for the individual for the sake of others and thus constitute a social dilemma, such as donation of food or other resources, others can be mutually beneficial, such as conflict resolution or coordination in a group, for example for hunting [24, 25].

Arguably, some kinds of strategic complexity have been considered in evolutionary games. Examples include strategy dependent stochastic transitions between social dilemmas [27, 28], deterministic transitions between games [29, 30], the dynamics of two or more evolutionary games played in parallel, in one shot [31–35] (also coined multigames [33]), or repeated interactions [36], heterogeneity in the payoff structure of the games played by the individuals [37–40], the interaction of social dilemmas with signaling games [41–43], or the interaction between different social dilemmas in interacting heterogenous public goods games [44, 45]. However, despite their overwhelming prevalence in human and possibly other animal populations, the strategic complexity resulting from the interaction of games with different structures has remained poorly studied in evolutionary games. Here, I consider such a strategically complex situation where individuals face different strategic settings, each with a different set of actions and outcomes. In the language of game theory, this is to say individuals play different games with different strategies and payoffs. In such a context, the individuals' strategic decisions in one game can depend on what happens in the other games. This provides a way to solve the self-referential problem in the models of indirect reciprocity, as it allows the reputation-building mechanism and the decision-making mechanism to occur at different levels, i.e., based on different strategic settings. As I show here, this observation can give rise to the evolution of a set of cooperation supporting moral norms.

In our model, individuals play a Prisoner's Dilemma, together with a second game, which I call game *B*, and is not necessarily a social dilemma. Individuals build a reputation based on their behavior in the Prisoner's Dilemma and act based on this reputation in the game *B*. I consider a situation where both games are played with the same opponent or when the two games are played with different opponents. When game *B* is a pure dominance game with only one Nash equilibrium, such as a Prisoner's Dilemma, the same problem incurred by indirect reciprocity models prevents the evolution of cooperation: Individuals are better off playing the Nash equilibrium irrespective of their opponents' reputation. Consequently, the Nash equilibrium of the two-stage game can be decomposed into two simple Nash equilibria of the two composing games. The situation changes when game *B* has more than one equilibrium such that individuals can coordinate on a superior equilibrium, or avoid coordination failure, by taking the information about the strategy of their opponent into account in their game B strategic choices. In this case, in addition to the trivial defective Nash equilibria, a cooperative equilibrium exists in which coupling between games emerges and individuals decide upon their strategy in the second game based on their opponent's strategy in the first game and cooperation in the social dilemma is sustained, although full cooperation often does not evolve. A static analysis of the games provides simple rules for the existence of a cooperative Nash equilibrium in the two-stage game. When game *B* is a coordination game, a cooperative pure strategy Nash equilibrium exists when the cost of cooperation is smaller than the excess payoff gained by coordinating on the superior equilibrium. This condition also ensures the existence of a cooperative fixed point in the evolutionary dynamics. When game *B* is a Snow Drift game, belonging to the anti-coordination class, a cooperative pure strategy Nash equilibrium exists when the cost of cooperation is smaller than the coordination asymmetry.

By examining the evolutionary dynamics when the game *B* is a Stag Hunt game, I show that when game *B* belongs to the coordination class, such that it has two symmetric equilibrium,

both a cooperative state where cooperators and defectors coexist, and a defective state, where defectors dominate are possible in a mixed population. However, the cooperative state has a small basin of attraction, and thus, convergence to such a state requires cooperation-favoring coordination norms to be encoded in the initial conditions. Furthermore, this cooperative state disappears in a structured population. The situation is different when game *B* belongs to the anti-coordination class, such that it possesses an asymmetric equilibrium. In this case, a symmetry-breaking phase transition exists above which the symmetry between cooperation and defection breaks. A set of behavioral rules emerges according to which cooperation stands out as a valuable trait, and individuals play softly with cooperators (i.e., they play strategies that gives a higher payoff to the opponent with cooperators). This leads to the evolution of cooperation. While the evolution of moral norms often lead to the coexistence of cooperative and defective strategies, in which cooperation maintained due to receiving a higher payoff from game *B*, depending on the strength of social dilemma and the structure of game *B*, moral norms can also fully suppress defection and give rise to full cooperation in the social dilemma in the stationary state of the dynamics. Importantly, this set of norms also promotes a more efficient allocation of resources in game *B*. This is particularly the reason why it evolves based on individuals' self-interest. This observation appears to conform to the view that many aspects of moral systems do not necessarily require self-sacrifice but simply help to foster mutualistic cooperation and bring order and organization into societies [46–50]. In this regard, our analysis suggests that moral systems behave like a Trojan horse: Once established out of the individual's self-interest, they also promote cooperation and self-sacrifice. Importantly, the conditions for the evolution of cooperation depend only on the cost of cooperation and are independent of other parameters of the Prisoner's Dilemma. The fact that the evolution of moral norms only depends on the cost of cooperation and is independent of its benefit implies that even bad norms, which incur a pure collective cost, can evolve and promote organization and order.

Analysis of the model in a structured population shows that population structure can remove the bistability of the system and ensures the evolution of cooperation favoring moral norms starting from all the initial population configurations when game *B* is an anti-coordination game. In contrast, in those simulations in which game B is a coordination game, we did not observe the evolution of cooperation. Furthermore, noise in inferring reputation can have a surprisingly positive effect on the evolution of a moral system in structured populations. However, this may not compensate for the loss cooperators experience due to a high recognition noise level. More ever, I show a very high level of recognition noise facilitates the evolution of cooperative behavior in the snow-drift game. This contrasts previous findings regarding the detrimental effect of population structure on the evolution of cooperation in the snow-drift game in simple strategic settings [51], and parallels some arguments regarding the beneficial effect of noise for biological functions [52–55].

The structure of the paper is as follows. First The model for both direct interactions and indirect interactions is introduced. Results Section begins by a Static analysis of the games and by deriving the Nash equilibria of the two-stage game shows that two classes of equilibria, the simple defective equilibria which are decomposable to two simple Nash equilibria of the composing games, and the cooperative equilibria where coupling between the games emerges, exists in the two-stage game, and derives simple Conditions for the existence of cooperative Nash equilibrium in terms of the cost of cooperation. Evolutionary dynamics: Mixed population studies the evolutionary dynamics in a mixed population and shows both defective fixed point where cooperation does not evolve and cooperative fixed points in which cooperation evolves exist. Fixed points of the evolutionary dynamics corresponding to the defective and cooperative equilibria are studied and by deriving the mixed strategy Nash equilibria

corresponding to the fixed points, it is shown that similarly to the conditions for the existence of the cooperative Nash equilibria, The evolution of moral norms depends only on the cost of cooperation. By studying the Basin of attraction of the fixed points it is shown that the basin of attraction of the fixed points are dramatically smaller when game *B* is a coordination game compared to when it is an anti-coordination game, which points towards fundamentally different mechanisms underlying the evolution of cooperation in these two cases. Then by studying the Time evolution of the system when game *B* belongs to the anti-coordination class, it is shown that a set of cooperation favoring moral norms evolve in the course of evolution through a rapid dynamical transition due to the density-dependent selection of a costly cooperative trait. Evolution of moral norms in the direct interaction model and Evolution of moral norms in the reputation-based model further study the evolution of moral norms in the direct interaction and reputation-based model by simulations in finite populations and replicator dynamics for different archetypal games belonging to the anti-coordination class and shows that moral norms can lead to both cooperation in the Prisoner's dilemma and a better allocation of resources and anti-coordination in game B. In Continuous variations of the structure of game *B* and the evolution of moral norms through a symmetry breaking phase transition by studying the continuous variation of the structure of game *B* the physics of the evolution of moral norms is studied and it is shown that moral norms evolve by a symmetry-breaking phase transition above which the symmetry between cooperation and defection breaks and a set of cooperation-favoring moral norms evolve. Finally, the model in a Structured population is studied and it is shown that while cooperation in a structured population does not evolve when game *B* is coordination game, it does evolve when game *B* is an anti-coordination game. Furthermore, it is shown that population structure removes the bistability of the system and ensures the evolution of moral norms starting from all the initial conditions. Finally, it is shown that recognition noise can facilitates the evolution of moral norms in a structured population. In the Discussion it is argued how the findings can shed light on the evolution of indirect reciprocity and the evolution of harmful norms, and how the mechanism underlying the evolution of moral norms relates to the evolution costly traits.

## The model

I begin by introducing two slightly different models. In the first model, the direct interaction model, information about the strategies of the individuals is acquired through direct observation. In this model, I consider a population of $N$ individuals. At each time step, individuals are randomly paired to interact. Each pair of individuals play a Prisoner's Dilemma (PD), followed by a second game. The second game is a two-person, two-strategy symmetric game, which I call game *B*. I call the two possible strategies of game *B*, down (d), and up (u) strategies. The strategy of an individual in game *B* is a function of its opponent's strategy in the first game. Thus, the strategy of an individual can be denoted by a sequence of three letters *abc*. Here, the first letter is the individual's strategy in the PD and can be either $C$ (cooperation) or $D$ (defection). The second letter is the individual's strategy in the game *B* if its opponent cooperates in the PD, and the last letter is the individual's strategy in the game *B* if its opponent defects in the PD. Clearly, we have $b, c \in \{u, d\}$. For example, a possible strategy is to cooperate in the PD, play *d* if the opponent cooperates, and play *u* if the opponent defects. I denote such a strategy by *Cdu*.

While in the direct interaction model, individuals play both their games with the same opponent, I also consider a second model, the reputation-based model, where individuals play their two games with different opponents. In this model, individuals are randomly paired to play a PD at each time step. After this, the interaction ends, and individuals meet another

randomly chosen individual to play their second game. Under this scenario, individuals do not observe their opponent's strategy in the PD. Instead, I assume individuals have a reputation of being cooperator or defector, on which their opponent's decision in the second game is based. For example, an individual with the strategy $Cdu$, cooperates in the PD, plays $d$ if it perceives its opponent to be a cooperator, and plays $u$ if it perceives its opponent to be a defector. To model reputation, I assume with a probability $1 − \eta$ individuals are informed about the PD-strategy of their opponent, and with probability $\eta$ they make an error in inferring the PD-strategy of their opponent. $\eta$ can be considered as a measure of noise in inferring the reputations. I note that, for $\eta = 0$, the dynamics of the two models are mathematically similar.

For the evolutionary dynamics, I assume individuals gather payoff according to the payoff structure of the games and reproduce with a probability proportional to their payoff. Offspring inherit the strategy of their parent. However, with probability $v$ a mutation occurs, in which case the strategy of the offspring is set to another randomly chosen strategy.

I will also consider a structured population. While in a mixed population, individuals randomly meet to interact, in a structured population, individuals reside on a network and interact with their neighbors. That is, each individual derives payoffs by playing its two games with all its neighbors. For the evolutionary dynamics, I consider an imitation rule, in which individuals update their strategy in each evolutionary step by imitating an individual's strategy in their extended neighborhood (composed of the individual and its neighbors) with a probability proportional to its payoff, subject to mutations. For the population network, I consider a first nearest neighbor square lattice with von Neumann connectivity and periodic boundaries.

The payoff values of the games are as follow. In the PD, individuals can either cooperate or defect. If both cooperate, both get a payoff $R$ (reward), and if both defect, both get a payoff $P$ (punishment). If an individual cooperates while its opponent defects, the cooperator gets a payoff $S$ (sucker's payoff), while its opponent gets a payoff $T$ (temptation). For a Prisoner's Dilemma, we have $S < P < R < T$ with $T + S < 2R$. For game $B$, I show the payoff of mutual down by $R_B$, and the payoff of mutual up by $P_B$. If an individual plays up while their opponent plays down, the up-player gets $T_B$, and the down-player gets $S_B$.

I will analyze the model for different structures for game $B$. Here I consider only symmetric games. Symmetric two-player two-strategy games can be divided into three classes: pure dominance (such as Prisoner's dilemma), coordination games (such as Stag Hunt game), and anti-coordination games (such as Snow Drift game) [56]. As mentioned in the introduction, when game $B$ is a pure dominance game, such as a social dilemma, cooperation in the two-stage game does not evolve, while cooperation can evolve when game $B$ is a coordination game or an anti-coordination game. As for the anti-coordination class, I consider cases where game $B$ is a Snow Drift (SD) game (also known as the Hawk-Dove or Chicken game) [2], the Battle of the Sexes (BS), and the Leader game. Together with the Prisoner's Dilemma, these games are coined as four archetypal two-person, two-strategy games [57]. I will also consider a case where game $B$ is a Stag Hunt (SH) game, which belongs to the coordination class [56]. Furthermore, I examine the dependence of the results on the continuous variation of the structure of the game $B$. All the anti-coordination games mentioned above have an asymmetric Nash equilibrium. In the Nash equilibrium, one of the strategies is superior in the sense that it leads to a higher payoff, leaving the opponent with a lower payoff. I call such a superior strategy the hard strategy, and the inferior strategy, which leads to a lower payoff in equilibrium, is called the soft strategy. When using the three archetypal games, I take the soft strategy to be the same as down and the hard strategy to be the same as up. The base payoff values used in this study (unless otherwise indicated) are presented in Table 1.

**Table 1. Base payoff values.**

|  | R | S | T | P |
|---|---|---|---|---|
| Prisoner's Dilemma | 3 | 0 | 5 | 1 |
|  | $R_B$ | $S_B$ | $T_B$ | $P_B$ |
| Snow Drift (SD) | 3 | 1 | 5 | 0 |
| Battle of the Sexes (BS) | 0 | 3 | 5 | 0 |
| Leader | 2 | 3 | 5 | 1 |
| Stag Hunt | 5 | 0 | 1 | 1 |

# Results

In the Results Section, I begin by a static analysis of the two-stage game followed by studying the evolutionary dynamics of both direct interaction model and reputation-based model in a well-mixed population, and end by studying the evolutionary dynamics in a structured population.

## Static analysis of the games

**Nash equilibria of the two-stage game.** I begin by deriving the Nash equilibria of the two-stage game in the direct interaction model. The two-stage game is composed of 8 possible strategies, namely, *Cuu, Cud, Cdu, Cdd, Duu, Dud, Ddu, Ddd*. The payoff matrix of the two-stage game can be read off based on the payoffs of the two composing games and is presented in Fig 1. Table 2 shows the numerical values of the payoffs when the game *B* is a Snow Drift (top) and a Stag Hunt (bottom) game. To see how to construct the payoff matrix of the two-stage game, as an example, consider the payoffs of strategies *Cud* and *Cdu* against each other. They both cooperate in the PD and receive a payoff of *R*. *Cud* plays up if its opponent cooperates, and *Cdu* plays down. Thus, *Cud* receives $T_B$ and *Cdu* receives $S_B$. The whole payoff matrix can be constructed in a similar way. The Nash equilibria of the two-stage game are strategy profiles, $(s, s')$ (*s* refers to the strategy of the row player and $s'$ to that of the column player), where none of the players has a unilateral incentive to change its strategy, i.e., nobody can increase its payoff by unilaterally switching to another strategy. The pure strategy Nash equilibria of the game are marked with colored cells in Table 2. When game *B* is an anti-

**Fig 1. The payoff matrix of the two-stage game.** The payoff matrix of the two-stage game can be constructed based on the payoffs of the first stage (prisoner's dilemma) and second stage games (game B). The payoffs of the row player are shown.

**Table 2. Payoff values of the two-stage game when game *B* is a Snow Drift game (top) and a Stag Hunt game (bottom).** The first number shows the payoff of the row player, and the second number shows the payoff of the column player. Cooperative Nash equilibria are indicated by green and defective equilibria by red cells.

| Game B: Snow Drift | | | | | | | |
|---|---|---|---|---|---|---|---|
| | *Cuu* | *Cud* | *Cdu* | *Cdd* | *Duu* | *Dud* | *Ddu* | *Ddd* |
| *Cuu* | 3,3 | 3,3 | 8,4 | 8,4 | 0,5 | 0,5 | 5,6 | 5,6 |
| *Cud* | 3,3 | 3,3 | 8,4 | 8,4 | 1,10 | 1,10 | 3,8 | 3,8 |
| *Cdu* | 4,8 | 4,8 | 6,6 | 6,6 | 0,5 | 0,5 | 5,6 | 5,6 |
| *Cdd* | 4,8 | 4,8 | 6,6 | 6,6 | 1,10 | 1,10 | 3,8 | 3,8 |
| *Duu* | 5,0 | 10,1 | 5,0 | 10,1 | 1,1 | 6,2 | 1,1 | 6,2 |
| *Dud* | 5,0 | 10, 1 | 5,0 | 10,1 | 2,6 | 4,4 | 2,6 | 4,4 |
| *Ddu* | 6,5 | 8,3 | 6,5 | 8,3 | 1,1 | 6,2 | 1,1 | 6,2 |
| *Ddd* | 6,5 | 8,3 | 6,5 | 8,3 | 2,6 | 4,4 | 2,6 | 4,4 |

| Game B: Stag Hunt | | | | | | | |
|---|---|---|---|---|---|---|---|
| | *Cuu* | *Cud* | *Cdu* | *Cdd* | *Duu* | *Dud* | *Ddu* | *Ddd* |
| *Cuu* | 4,4 | 4,4 | 4,3 | 4,3 | 1,6 | 1,6 | 1,5 | 1,5 |
| *Cud* | 4,4 | 4,4 | 4,3 | 4,3 | 0,6 | 0,6 | 5,10 | 5,10 |
| *Cdu* | 3,4 | 3,4 | 8,8 | 8,8 | 1,6 | 1,6 | 1,5 | 1,5 |
| *Cdd* | 3,4 | 3,4 | 8,8 | 8,8 | 0,6 | 0,6 | 5,10 | 5,10 |
| *Duu* | 6,1 | 6,0 | 6,1 | 6,0 | 2,2 | 2,1 | 2,2 | 2,1 |
| *Dud* | 6,1 | 6,0 | 6,1 | 6,0 | 1,2 | 6,6 | 1,2 | 6,6 |
| *Ddu* | 5,1 | 10,5 | 5,1 | 10,5 | 2,2 | 2,1 | 2,2 | 2,1 |
| *Ddd* | 5,1 | 10,5 | 5,1 | 10,5 | 1,2 | 6,6 | 1,2 | 6,6 |

coordination game (SD), the two-stage game has two classes of Nash equilibria. In one set of equilibria, which I call the defective equilibria, the two games remain decoupled, and in the Nash equilibrium of the two-stage game, the Nash equilibrium of the composing games is played in each stage. Thus, in the PD, mutual defection is played, and in the SD, the strategy pair down-up is played. Consequently, cooperation is not supported in this equilibrium.

In another equilibrium, denoted by green cells, the two games become coupled: The strategy in game *B* depends on the opponent's strategy in the first stage. Consequently, the Nash equilibrium of the two-stage game can not be decomposed into two simple Nash equilibrium. In this equilibrium, players use information about the strategy of their opponent in the first stage to coordinate on a heterogeneous strategy pair and avoid a coordination failure. Consequently, cooperation in the PD is supported based on the fact that it provides a mechanism for the agents to avoid a coordination failure. By studying other classes of anti-coordination games (see the Supporting Information Text, section 3), I show that the same picture holds for other cases of anti-coordination game.

I note that an interesting feature of the cooperative Nash equilibrium is that defectors play soft strategy with cooperators in equilibrium, and cooperators play hard strategy with defectors. This ensures cooperators receive a higher payoff from game *B* in the cooperative equilibrium, compensating for the cost of cooperation they pay. As I show shortly, this feature underlies the survival of cooperators in the evolutionary games.

The case in which game *B* is a coordination game is considered in the bottom table in Table 2. Here, game *B* is a Stag Hunt game, which has two equilibria. In the superior equilibrium, players coordinate on *d*, and in the inferior equilibrium, they coordinate on *u*. The defective Nash equilibria of the two-stage game are composed of four equilibria in which players defect in the PD and coordinate on the superior strategy, *d*, in the coordination game, and

one equilibrium in which players defect in PD and coordinate on the inferior strategy in the coordination game.

Cooperative equilibria are composed of two classes as well. In one class, mutual cooperation in the PD occurs. In this equilibrium, both players play *Cdu*: They cooperate in the PD, play *d* with cooperators and play *u* with defectors. In four Nash equilibria, players coordinate on heterogeneous cooperation and defection pairs. In these equilibria, the defector plays *d* only if its opponent cooperates. Thus as long as a cooperator plays *d* with a defector, this strategy profile remains a Nash equilibrium. In both classes of cooperative equilibrium, when game *B* is a coordination game, cooperation survives due to the existence of cooperation favoring coordination rules.

**Conditions for the existence of cooperative Nash equilibrium.** It is possible to derive simple rules for the existence of cooperative Nash equilibrium. I begin by the cooperative Nash equilibrium of the snow drift game, (*Cuu*, *Ddu*). Using the general parametrization of the Snow Drift game ($T_B > R_B > S_B > P_B$), from the condition $BR(Cuu) = Ddu$ we arrive at $T + S_B > R + S_B$, $T + S_B > T + P_B$, and $T + S_B > R + P_B$ which always hold for SH (BR(x) stands for best response to x). Using the condition $BR(Ddu) = Cuu$, we arrive at, $S + T_B > S + R_B$, $S + T_B > P + P_B$, and $S + T_B > P + S_B$. The first inequality is automatically satisfied, and the second one is satisfied when the last one is satisfied. The last inequality gives $P - S < T_B - S_B$. If this condition is violated, no other pure strategy Nash equilibrium involving cooperative strategies exists. To see this, I note that *Ddd* dominates all the cooperative strategies against defective strategies. So none of the cooperative strategies can be the best response to any of the defective strategies. Thus, the only remaining possibility for the existence of a cooperative pure strategy Nash equilibrium is that a fully cooperative Nash equilibrium exists, which can not happen since the best response to any of the cooperative strategies is one of the defective strategies.

Using a similar argument for the Stag Hunt game, it is easy to see that the condition for the existence of a cooperative Nash equilibrium, in this case, is $P - S < R_B - P_B$ (see Fig 2). These two conditions can be stated as simple rules for the existence of a cooperative Nash equilibrium. To see this, I note that $P - S$ can be considered as the cost of cooperation. To see this consider the helping game version of the Prisoner's dilemma, in which the cooperator incurs a cost *c* for its opponent to receive a benefit *b*. In this version we have $R = b - c$, $T = b$, $P = 0$, $S = -c$. Thus, $P - S = c$. In the Snow Drift game, $T_B - S_B$ can be considered as coordination asymmetry, as this is the excess payoff of the superior strategy in the Nash equilibrium. Thus, when game *B* is a Snow Drift game, a cooperative Nash equilibrium exists when the cost of cooperation is smaller than the coordination asymmetry. In the Stag Hunt game, $R_B - P_B$ is the payoff difference of players in the superior and inferior Nash equilibria, and can be considered as the cost of coordinating on an inferior equilibrium. Thus, the condition for the existence of a cooperative Nash equilibrium when game *B* in a Stag Hunt game is that the cost of cooperation should be smaller than the cost of coordinating on an inferior equilibrium. As shown in the next section, this condition also ensures the existence of a cooperative fixed point in the evolutionary dynamics.

## Evolutionary dynamics: Mixed population

**Fixed points of the evolutionary dynamics.** An static analysis of the game suggests coupling between the two games can give rise to non-trivial cooperative equilibria where deviation from the Nash equilibrium of the simple games is observed. This raises the question of whether the evolutionary dynamics of interacting games can also give rise to cooperative fixed points? In this section we show this is the case and the evolutionary dynamics possesses both cooperative and defective fixed points. As shown in the Methods section, the dynamics of the model

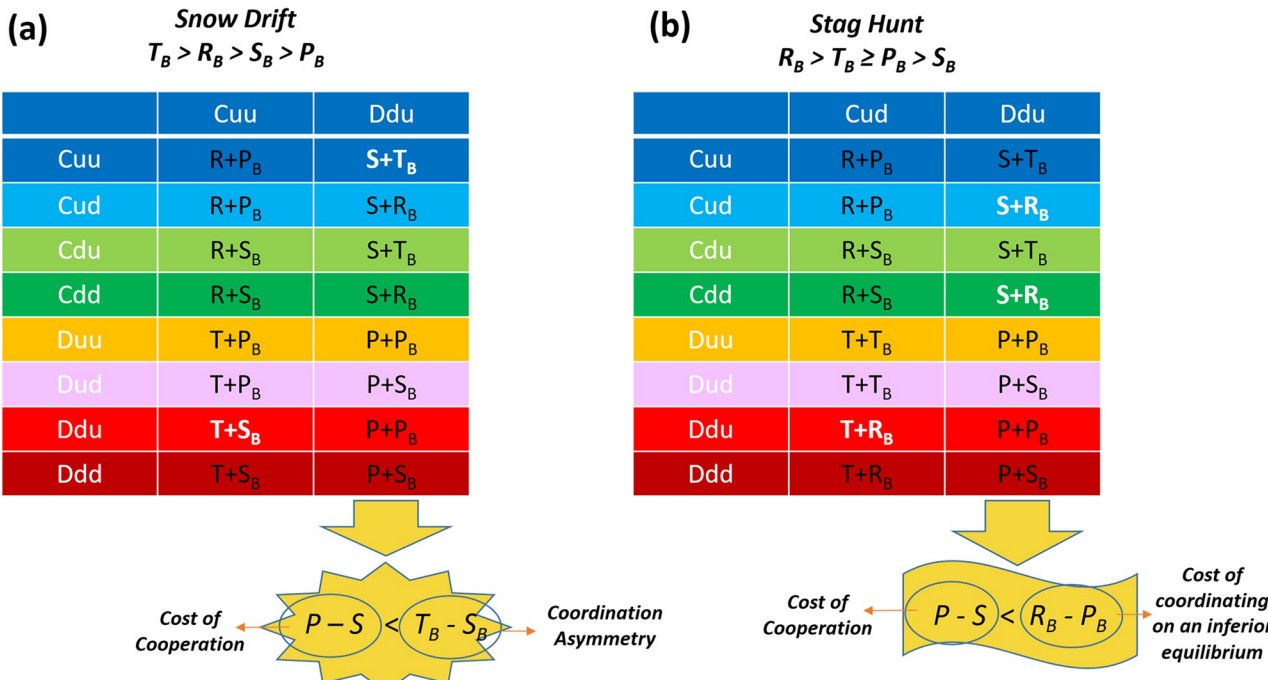

**Fig 2. Simple rules for the existence of a cooperative Nash equilibrium.** A: A cooperative Nash equilibrium in which defectors play softly with cooperators and cooperators play hard with defectors when the game $B$ is a Snow Drift game exists when the cost of cooperation is smaller than the coordination asymmetry defined as the payoff difference of hard and soft strategies in the Snow Drift game. B: A cooperative Nash equilibrium, when the game $B$ is a Stag Hunt game, exists when the cost of cooperation is smaller than the cost of coordinating on an inferior equilibrium, defined as the payoff difference of the superior and inferior equilibria in the Stag Hunt game. These conditions can be derived by requiring the payoff of Nash strategies (shown in white) to be larger than all the other payoffs in the same column.

can be described in terms of the replicator-mutator dynamics. Analysis of the replicator-mutator dynamics shows that the dynamics can settle in different fixed points. When game $B$ is an anti-coordination game, two fixed points are observed. A cooperative fixed point in which cooperators survive and a defective fixed point where defectors dominate. In Fig 3A to 3C the frequency of different strategies in the fixed points of the dynamics in the cases that the game $B$ is respectively, SD, BS, and the Leader game are plotted. In the defective equilibrium, all the defective strategies are found in a high proportion in the population (this is not the case in the BS). In this fixed point, as all the agents play the same strategy in the PD, they can not use the information from the PD to coordinate in game $B$. On the other hand, in the cooperative fixed point, cooperators and defectors can use information about their opponent strategy in the PD to coordinate on heterogeneous strategy pairs. Interestingly, in this fixed point, defectors always play the soft strategy with cooperators, and cooperators play the hard strategy with defectors. This ensures cooperators reach a higher payoff in game $B$, which compensates for the cost of cooperation they pay in the PD and sustain cooperation in the population. On the other hand, cooperators and defectors among themselves play a combination of soft and hard strategies.

The situation is different when game $B$ is the Stag Hunt game. In this case, the replicator-mutator dynamics have three stable fixed points. The frequency of strategies in the fixed points is plotted in Fig 3D. In one of the fixed points, only those defective strategies which play down with defectors survive. This corresponds to the superior defective Nash equilibrium with the payoff of $P + R_B$. On the other hand, the inferior Nash equilibrium where agents play up in the

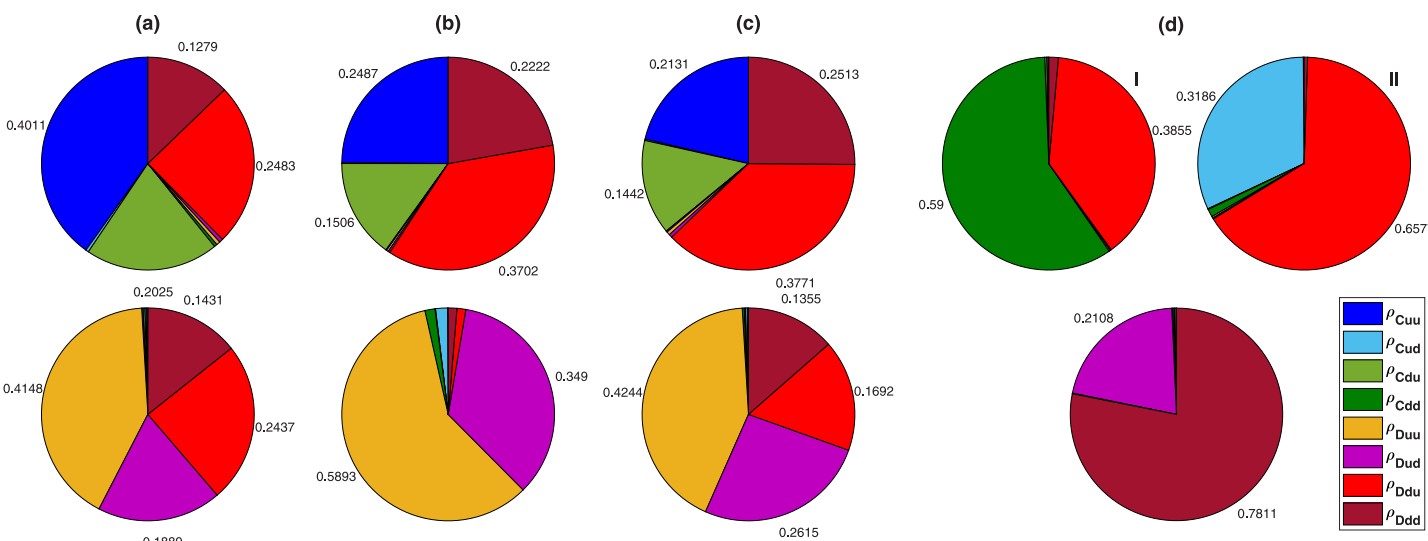

**Fig 3. Frequency of different strategies in the fixed points of the replicator-mutator dynamics.** Form A to D the game *B* is respectively, Snow Drift, Battle of the Sexes, Leader, and Stag hunt game. For anti-coordination games (A to C) the replicator-mutator dynamic has two stable fixed points, a cooperative fixed point where cooperation evolves (top) and a defective fixed point where cooperation does not evolve (bottom). For the Stag Hunt game, D, the replicator-mutator dynamic has two cooperative fixed points (top) and a defective fixed point (bottom).

Stag Hunt game, *Duu*, is not evolutionarily stable as mutant *Dud* receives the same payoff as *Duu*s in their presence and can grow in a population of homogeneous *Duu* players. This gives an edge to cooperators as cooperators can receive a higher payoff and invade by coordinating on the higher payoff *d* − *d* pair with mutant *Dud*s.

The two cooperative fixed points when game *B* is SH correspond to cases where the population is composed of *Cdd* − *Dud* or *Cud* − *Dud*. I call these fixed points, respectively, $SH_1$ and $SH_2$. Both these compositions correspond to the Nash equilibrium of the two-stage game. The Nash equilibrium *Cdu* − *Cdu*, on the other hand, is not evolutionarily stable, as *Cdd* players can grow in such a homogeneous population. Once they grow, defective strategies can invade the population due to receiving a high payoff in the presence of *Cdd* strategies.

In passing, I note that, while cooperative fixed points are possible in both the cases that game *B* is an anti-coordination game and when it is a coordination game, as we will shortly see, the mechanisms underlying these stationary states are rather different, and the likelihood that these fixed points occur in a dynamical process are dramatically different.

**The evolution of moral norms depends only on the cost of cooperation.** An interesting question is that whether, similarly to the simple conditions for the xistence of cooperative Nash equilibria, does conditions for the existence of cooperative fixed points in the evolutionary dynamics exist? The fixed points of the evolutionary dynamics correspond to mixed strategy Nash equilibria of the static game (see Methods). Using the general parametrizations of the payoffs, the fixed point $SH_1$ corresponds to a mixed strategy Nash equilibrium in which the strategies *Cdd* and *Dud* are played with probabilities, respectively, $x_{Cdd} = \frac{P - R_B - S + S_B}{P + R - R_B - S + S_B - T}$ and $x_{Dud} = \frac{R - T}{P + R - R_B - S + S_B - T}$, and all the other strategies are played with probability zero. The fixed point $SH_2$ corresponds to a mixed strategy Nash equilibrium in which *Cud* and *Dud* are played with probabilities, respectively, $x_{Cud} = \frac{P + P_B - R_B - S}{P + 2P_B + R - 2R_B - S - T}$ and $x_{Ddu} = \frac{P_B + R - R_B - T}{P + 2P_B + R - 2R_B - S - T}$. These expressions describe the fixed point of the dynamics for coordination class in the limit of zero mutation rate. It is possible to derive simple conditions for the evolution of cooperation using these

expressions. To do this, I note that an interesting feature of the fixed points is that they only depend on two combinations, $P - S$ and $T - R$. Using a helping game version of the Prisoner's dilemma ($R = b - c$, $T = b$, $P = 0$, $S = -c$) both these expressions are equal to the cost of cooperation: $P - S = T - R = c$. That is, the condition for the existence of a cooperative fixed point can be stated in terms of an inequality for the cost of cooperation. Using this parametrization, the equilibrium frequencies for the fixed point $SH_1$ can be written as $x_{Cdd} = 1 - c/(R_B - S_B)$ and $x_{Dud} = c/(R_B - S_B)$. Thus, this fixed point exists as long as $c < R_B - S_B$. For $SH_2$ we have, $x_{Cud} = [1 + c/(R_B - P_B)]/2$ and $x_{Dud} = [1 - c/(R_B - P_B)]/2$. This fixed point exists as long as $c < R_B - P_B$. For the Stag Hunt game, we have $R_B > T_B \geq P_B > S_B$. Thus, both inequalities are satisfied when the cost of cooperation is smaller than the cost of coordinating on an inferior equilibrium, $R_B - P_B$.

The mixed strategy Nash equilibrium composed of the strategies $Cuu$, $Cdu$, $Ddu$, and $Ddd$, corresponding to the fixed point of the evolutionary dynamics when game $B$ is an anti-coordination game is derived in the Methods section. In this case, too, the fixed points of the evolutionary dynamics depend on the payoff values of the PD only through the cost of cooperation. This implies that cooperation evolves when the cost of cooperation is smaller than a value determined by the structure of game $B$. In Fig 4, I plot the mixed strategy Nash equilibrium, together with the cooperative fixed point of the evolutionary dynamics for two different mutation rates and benefit of cooperation as a function of the cost of cooperation $c$. Here, a helping game version of the Prisoner's Dilemma is used. The game $B$ is a Snow Drift game with the base payoff values. As can be seen, the mixed strategy Nash equilibrium involving $Cuu$, $Cdu$, $Ddu$, and $Ddd$ exist for small enough costs and coincides with the fixed point. For larger cost, the cooperative mixed strategy Nash equilibrium does not exist. The dynamics becomes

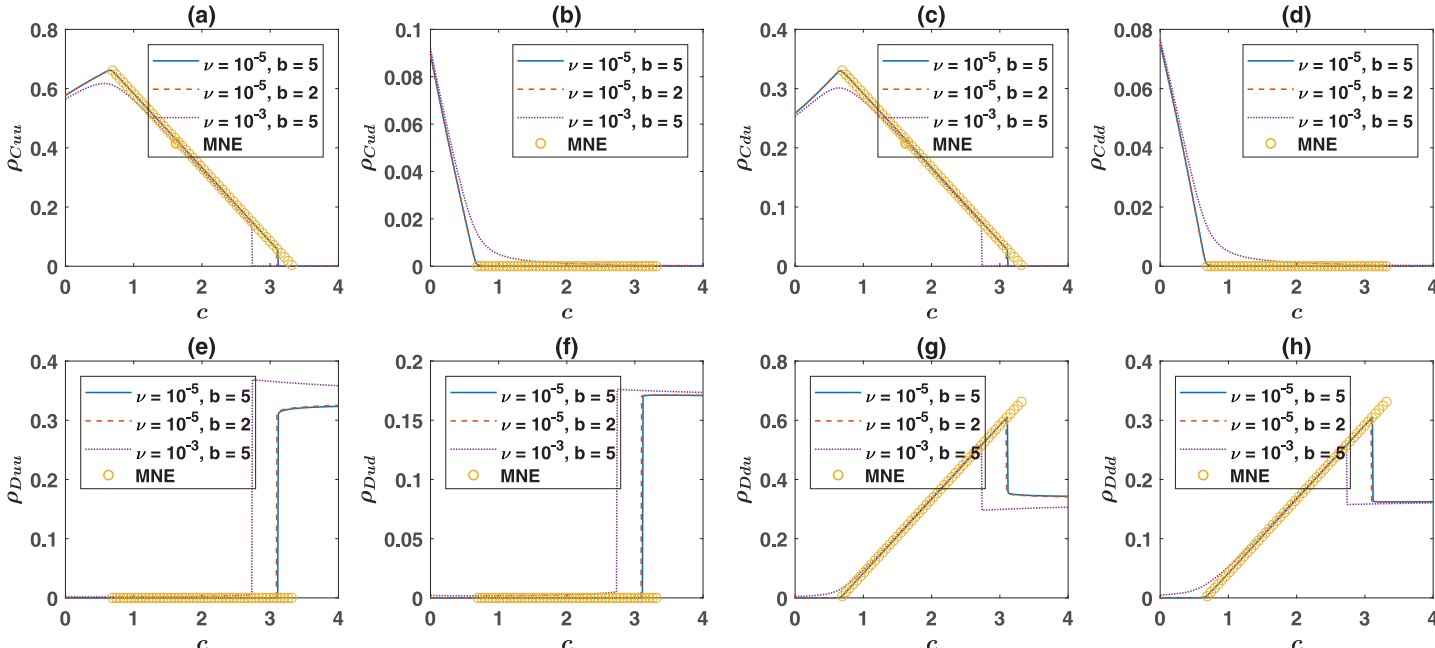

**Fig 4. The evolution of cooperation depends only on the cost of cooperation.** The cooperative fixed point of the evolutionary dynamics when the game $B$ is a Snow Drift game is plotted for two different mutation rates and two different benefits of cooperation (as indicated in the legend). The mixed strategy equilibrium (MNE) composed of $Cuu$, $Cdu$, $Ddu$, and $Ddd$, which coincides with the cooperative fixed point for zero mutation rate, is also plotted (orange circles). A cooperative fixed point exists for small costs and becomes unstable for high costs. The dynamics settle in a fully cooperative fixed point for too small costs where only cooperative strategies survive. Here a helping game version of the Prisoner's Dilemma with payoff values $R = b - c$, $T = b$, $P = 0$, and $S = -c$ is used. A base payoff of $\pi_0 = 5$ is added to all the individuals. For the Snow Drift game, the base payoff values, presented in Table 1, is used.

 

mono-stable and only a defective fixed point exist. This defective fixed point in turn corresponds to a defective mixed strategy Nash equilibrium of the two-stage game, which is decomposable to the Nash equilibrium of the PD and the mixed strategy Nash equilibrium of the SD. In other words, the frequency of up and down strategies in this case obey their frequency in the mixed strategy Nash equilibrium of a simple Snow Drift game.

There is another transition for smaller cost of cooperation below which the cooperative mixed strategy Nash equilibrium does not exist. In this region, the dynamics settle in a fully cooperative fixed point, in which only cooperative strategies survive. This fully cooperative fixed point corresponds to a fully cooperative mixed strategy Nash equilibrium, in which only cooperative strategies, *Cuu*, *Cud*, *Cdu*, and *Cdd* are played with non-zero probability. This mixed strategy Nash equilibrium is derived in the Methods section. It is also possible to derive a condition for the existence of the fully cooperative fixed point. For the payoff values of the Snow Drift game used here, a fully cooperative fixed point exists when the cost of cooperation is smaller than 2/3. As we will shortly see, this fixed point does not exist for other anti-coordination games considered here. The existence of such a fully cooperative fixed point might seem like a puzzle, as one might wonder that the advantage of cooperators in achieving a higher status in the anti-coordination task in the second round perishes in the absence of defectors. As we will shortly see, this puzzle can be solved by looking into the dynamics of the system.

**Basin of attraction of the fixed points.** So far, we have seen the dynamics can settle in different fixed points. An interesting question is, what is the likelihood that these fixed points occur starting from different initial population configurations? In this section, I show that the basin of attraction of the cooperative fixed points when game *B* is an anticoordination game is significantly larger than when it is a coordination game, which points towards different evolutionary processes underlying the evolution of cooperative norms in these two cases. To see this, in Fig 5A, I plot the probability that the dynamics settle in the cooperative fixed points for different structures of game *B*. To calculate these probabilities, I solve the replicator-mutator dynamics using $10^7$ randomly generated initial conditions. The results show that while for the anti-coordination class, the cooperative fixed point has a large basin of attraction and can occur for a broad range of initial conditions, both cooperative fixed points when game *B* is the Stag Hunt game, $SH_1$ and $SH_2$, occur for a narrow range of initial conditions. For instance,

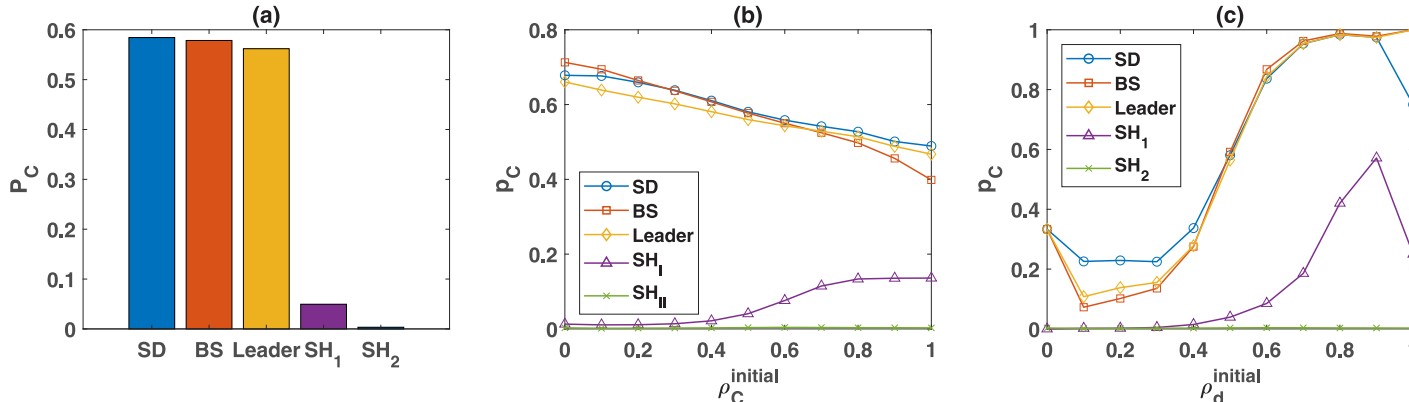

**Fig 5. The probability of settling in the cooperative fixed point.** A: The probability of settling in cooperative fixed point starting from random initial conditions for different structures of game *B*. While for anti-coordination games, the cooperative fixed point has a large basin of attraction, for the Stag Hunt game, cooperative fixed points can occur only for special initial conditions. B to C: The probability of settling in the cooperative fixed point as a function of the initial frequency of cooperators $\rho_C^{initial}$, A, and initial frequency of soft players, $\rho_d^{initial}$, B, for different structures of game *B* is plotted. The replicator-mutator dynamic is solved for $10^7$ different randomly generated initial conditions to derive the probabilities.

these fixed points do not occur for an unbiased initial condition in which the density of all the strategies is equal.

The difference between the basin of attraction of the two classes of the games points to the difference in the mechanism underlying the evolutionary processes leading to these stationary states. In the case of the anti-coordination game, individuals are unable to avoid an anti-coordination failure in the defective fixed point, while they can more effectively avoid such failure in the cooperative fixed point and coordinate on heterogeneous strategy pairs in game $B$ by taking the PD-strategy of their opponent into account. As this anti-coordination is beneficial for cooperators and defectors alike, these fixed points can occur due to the individuals' attempt to maximize their payoff by coordination on heterogeneous strategy pairs. Consequently, they have a relatively large basin of attraction and can occur starting from rather diverse initial conditions.

On the other hand, when game $B$ is a Stag Hunt game, individuals can do equally good in a heterogeneous population of cooperators and defectors (i.e., in a cooperative fixed point) or a homogeneous population of defectors (i.e., in the defective fixed point) in avoiding a coordination failure, provided a proper norm of coordination exists (in the former case, such a norm amounts to cooperators playing $d$ with defectors and defectors playing $d$ with cooperators and in the latter case it amounts to defectors playing $d$ with fellow defectors). Consequently, converging to a cooperation-favoring norm of coordination does not bring any direct advantage for the individuals. Rather, it can only occur if the initial population configuration is prepared in such a way to favor these norms.

To see what characteristics of the initial population configurations affect the likelihood of settling in a cooperative fixed point, I consider the projections of the eight-dimensional simplex defining the phase space of the system into a one-dimensional space defined by $\rho_C$ and $\rho_d$, respectively, in Fig 5B and 5C. In these figures, I plot the probability of settling into the cooperative fixed points, starting from randomly chosen initial conditions for different structures of game $B$ as a function of, respectively, the initial frequency of cooperators, $\rho_C^{initial}$, and the initial frequency of soft players, $\rho_d^{initial}$. To calculate these plots, I numerically solve the replicator-mutator dynamics using $10^7$ randomly generated initial conditions and derive the probability of settling into the cooperative fixed point, given the initial frequency of cooperators (Fig 5B) or soft players (Fig 5C) based on this sample.

When game $B$ is an anti-coordination game, converging to the cooperative state is favored when the initial frequency of cooperators is low. This originates from the fact that when the initial frequency of cooperators is low, strategies which play softly with cooperators bear a lower fitness cost on their bearer and grow faster. Consequently, cooperation-favoring moral norms are easier to evolve for such initial configurations. The situation is different when game $B$ is a coordination game. In this case, the dynamics have two cooperative fixed points, both with a significantly smaller basin of attraction. As can be seen in Fig 5B, the fixed point with lower cooperation, $SH_2$, composed of $Cud$ and $Ddu$, although stable, has a very small basin of attraction and the one with higher cooperation, $SH_1$, composed mainly of $Cdd$ and $Dud$ strategies, can occur for larger values of $\rho_C$. This is the case because when the initial frequency of cooperators is higher, a $Cdd$ strategy meets cooperators with higher probability and has more chances to reach a higher payoff due to coordination in game $B$.

The dependence of the probability of the occurrence of fixed points on the initial frequency of soft players, $\rho_d^{initial}$, shows for all the structures of game $B$, the cooperative fixed point is more likely to occur when the initial frequency of down strategy is higher, which shows that the evolution of cooperation-supporting norms is easier in populations composed of a higher frequency of soft players.

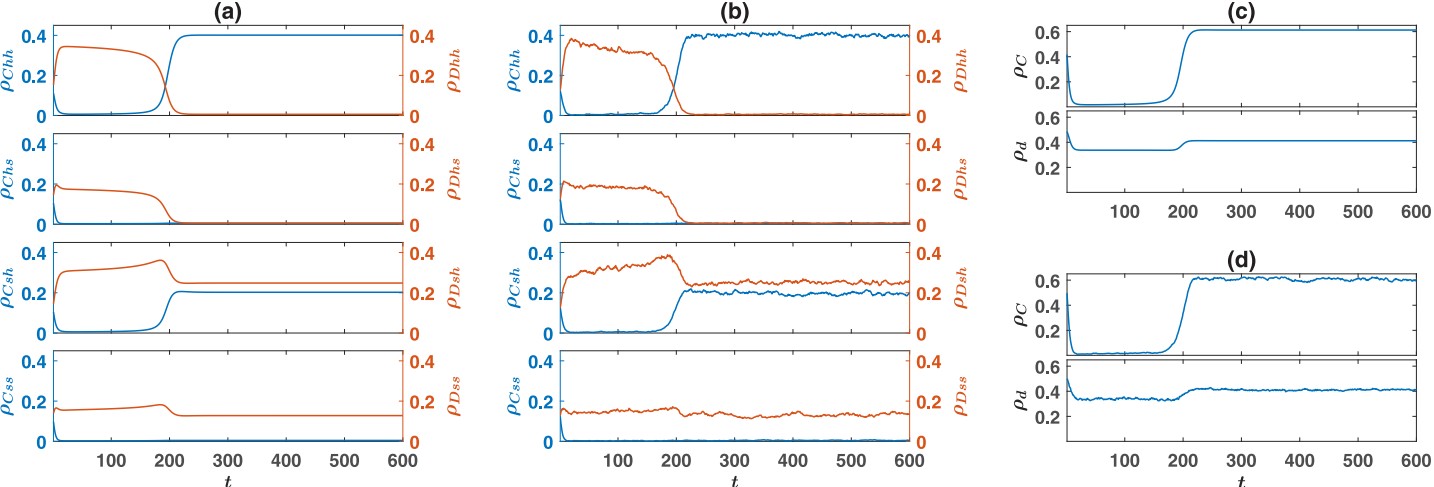

**Fig 6. Time evolution in the model with direct interactions.** A and B: The time evolution of different strategies resulted from the replicator-mutator dynamics A and a simulation in a finite population B. (c) and (d): The time evolution of the density of the cooperators $\rho_C$ (up), and the density of the soft strategies $\rho_d$ (bottom), resulted from the replicator-mutator dynamics C, and a simulation D. Cooperation favoring moral norms evolves through a rapid dynamical transition. The simulation is performed on a population of size $N = 20000$, and the mutation rate is $\nu = 0.005$. The initial condition is a random assignment of strategies (for the replicator-mutator dynamics, this implies $\rho_x = 1/8$, for all strategies $x$). The base payoff values, presented in Table 1, are used.

**Time evolution of the system.** As we have seen, when game *B* is an anti-coordination game, starting from a rather broad range of initial conditions, a set of cooperation-supporting moral norms emerge through the dynamics and help the maintenance of cooperators in the population. To see the nature of the dynamical phenomena through which this happens, in Fig 6A and 6B, I plot the density of different strategies, in the direct interaction model, as a function of time. Here, game *B* is a Snow Drift game. The dynamic is similar for other anti-coordination games, and in the reputation-based model. As can be seen in Fig 6C (replicator-mutator dynamics) and Fig 6D (simulations in a population of size 10000), where the density of cooperators in the PD and soft strategy in the SD are plotted, starting from the center initial condition in which all the strategies are found in the same density, the population rapidly goes to a state where the density of strategies in both games is close to its Nash equilibrium value. This is the case because, initially, no coupling between the two games exists, and strategies of the individuals evolve in such way that the frequency of strategies in each game follows its Nash equilibrium value. Consequently, the densities of all the strategies are close to their value in the defective fixed point, such that it might appear that the system has settled in the defective phase. However, this only sets the stage for the second phase of the evolution, during which a set of cooperation-supporting moral norms evolve and give rise to an outburst of cooperation. As cooperators are found in a very small frequency during this transient phase, strategies that play softly with cooperators do not impose a high cost on their bearer and grow in number. A cooperation favoring moral system is established when such strategies accumulate enough. At this point, the system shows a rapid dynamical transition to the cooperative fixed point where cooperators emerge in large numbers. In this regime, cooperators always play a hard strategy with defectors, and defectors always play the soft strategy with cooperators. This compensates for the cost of cooperation that they pay. On the other hand, both cooperators and defectors play a combination of soft and hard strategies among themselves. This phenomenology shows, when a game has an asymmetric equilibrium, as is the case in anti-coordination games, individuals can use information about the strategy of their opponent in a social dilemma to efficiently coordinate in an asymmetric equilibrium and avoid paying the cost of coordination

failure. Consequently, a set of behavioral or moral norms emerges, according to which cooperators are allowed to play hard and deserve to be played soft with. This supports cooperation by compensating for the cost of cooperation. Importantly, by facilitating anti-coordination, this mechanism also increases the cooperative behavior in the second game.

I note that when game *B* is a Stag Hunt game, the time evolution of the system shows different behavior. In this case, a cooperative fixed point can occur only for special initial conditions chosen close enough to the cooperative fixed points. Consequently, the two dynamical phases and the rapid dynamical transition observed in the time evolution of the system when game *B* is an anti-coordination game is not at work in this case. Since when game *B* is a coordination game, a cooperative fixed point can occur only for special initial conditions, in the remainder of this section, we focus on anti-coordination games.

**Evolution of moral norms in the direct interaction model.** To more closely see how cooperators survive in the dynamics, I set $S = 0$, $P = 1$, and $R = 3$, and plot the frequency of cooperators in Fig 7A, and the density of the soft strategies in Fig 7B, as a function of $T$. Here, from top to bottom, the second game is the SD, the BS, and the Leader game. In each panel, I present the result of simulations (marker), together with the numerical solutions of the replicator-mutator dynamics (lines). The replicator-mutator dynamics show the model is bistable: Depending on the initial conditions, two fixed points, each with a high or a low level of cooperation are possible. The two fixed points of the dynamics are plotted by solid and dashed lines.

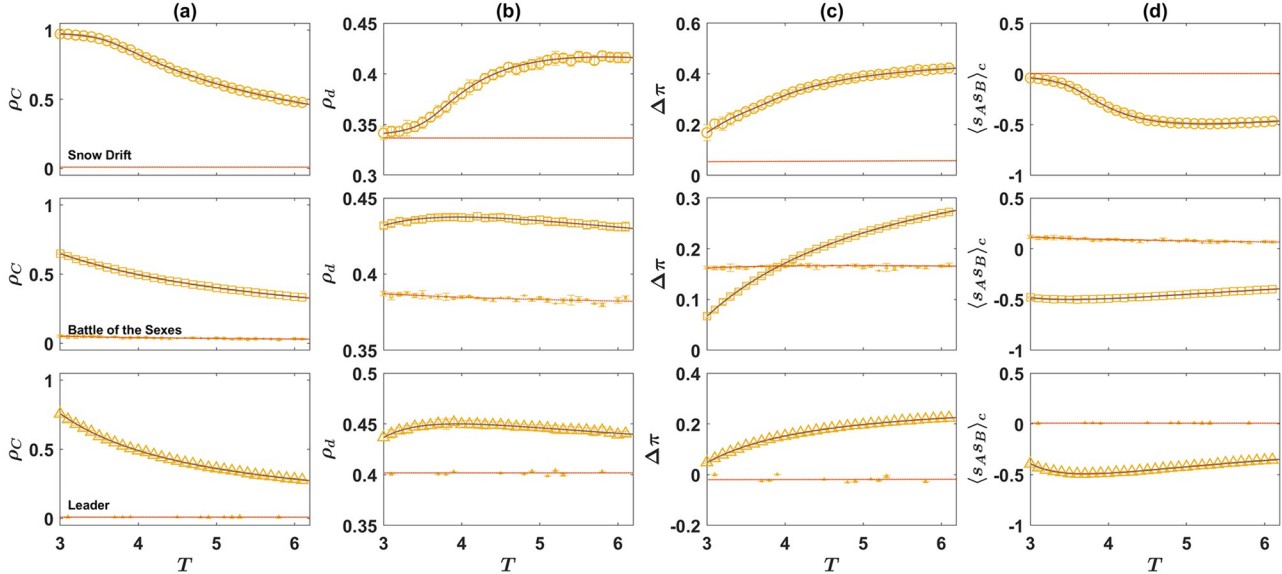

**Fig 7. The direct interaction model with the three archetypal games.** The density of cooperators, $\rho_C$, A, the density of soft strategies in game $B$, $\rho_d$, B, the normalized payoff difference of cooperators and defectors in game $B$, $\Delta\pi$, (c), and the correlation between the strategy of the individuals in the two games, $\langle s_A s_B \rangle_c$, D, as a function of the temptation, $T$. Here, from top to bottom, the game $B$ is the Snow Drift, the Battle of the Sexes, and the Leader game. The payoff values used for the games are presented in Table 1. The lines show the result of the replicator-mutator dynamics, and the markers show the results of simulations. The solid blue line shows the equilibrium fixed point, which occurs starting from an unbiased initial condition in which the density of all the strategies are equal, and the dashed red line shows the non-equilibrium fixed point, which can occur starting from certain initial conditions. The system is bistable and both a cooperative fixed point with a high level of cooperation A and soft strategies B, and a defective fixed point with a low level of cooperation and soft strategies are possible. In the cooperative phase, cooperators receive a higher payoff from game $B$ C. Moreover, the strategies of individuals show an anti-correlation in the cooperative fixed point D, resulting from the fact that defectors play softly with cooperators and cooperators play hard with defectors in this fixed point. For the simulations, a sample of 80 simulations, in a population of size $N = 10000$ is used. The simulations start from random initial conditions. In each simulation, the dynamics settle in one of the two fixed points. The markers show the averages, and the error bars show the standard deviation in the sample of simulations that settle in the given fixed point, and the size of markers is proportional to the number of times that the given fixed point occurs. Here, the mutation rate, $v = 0.005$. The simulations are run for 20000 time steps, and an average over the last 1000 time steps is taken.

The solid line represents the stationary state of the dynamics, starting from a center (unbiased) initial condition in which the density of all the strategies is equal. This can be considered as the unbiased fixed point [58]. The dashed line represents the biased fixed point, which can occur for certain initial conditions.

As can be seen, the replicator-mutator dynamics predict starting from a center initial condition, the system settles in the cooperative fixed point for all the values of $T$. While this is often the case for a simulation in a finite population, depending on the structure of game $B$, with a low probability, the biased fixed point, where a low level of cooperation is observed, can occur. This is the case for the BS game. The density of soft strategies, plotted in Fig 7B, is higher in the cooperative fixed point as well. On the other hand, in the non-cooperative fixed point, the density of soft strategies is smaller and close to the Nash equilibrium of the corresponding games. As mentioned before, this shows that coupling between games not only promotes cooperation in the PD, but also increases cooperative behavior in game $B$. I note that soft strategies, such as the strategy down in the Snow Drift game, can be considered as cooperative strategies as they allow their opponent to receive a higher payoff and thus benefit the opponent. However, in contrast to cooperation in Prisoner's Dilemma, soft strategies can occur in the Nash equilibrium and can be thought of as a more rational and self-interested form of cooperation [2]. In this sense, by increasing the frequency of soft strategies in the game $B$, moral norms also promote the less self-sacrificing and more self-centered form of cooperation.

The examination of the SD game in the top panel of Fig 7A shows that for small temptation, $T$, cooperators reach a frequency close to one. This might seem counter-intuitive as, in this case, since the population is homogeneous, the PD-strategy can not be used as a coordination device to anti-coordinate on heterogeneous strategy pairs in the game $B$. The key to the answer to the puzzle is the observation that the essential factor for the evolution of cooperation-favoring moral norms is that cooperative strategies reach a low frequency during the initial transient dynamic of the system before a set of moral norms emerge (see Fig 7). This transient dynamic guarantees the evolution of moral norms. Once the cooperative norms are fixated, cooperators can indeed increase and dominate the population.

As cooperators receive a lower benefit from the social dilemma, they can survive only if their payoff from game B compensates for the cost of cooperation they pay in the social dilemma. To see this is indeed the case, in Fig 7C, I plot the normalized payoff difference of cooperators and defectors in the game $B$, $\Delta\pi = (\pi_C^B - \pi_D^B)/(\pi_C^B + \pi_D^B)$. In all the three games, in the cooperative fixed point, cooperators receive a higher payoff in game $B$. In the SD and the BS games, cooperators receive a higher payoff in game $B$, even in the non-cooperative fixed point. As mentioned before, the higher payoff of cooperators from game $B$ results from the fact that in the course of evolution, individuals develop strategies that tend to play soft with cooperators and hard with defectors. This can be considered as the emergence of a set of moral rules that allows cooperators to reach a benefit by being treated softly. Consequently, cooperators can reach a higher payoff by playing a hard strategy. Besides, by increasing the strength of the social dilemma (that is, by increasing the temptation $T$), the payoff difference in the game $B$ increases in favor of the cooperators. This is due to an increase in the likelihood that cooperators receive a soft encounter in the game $B$, by increasing the strength of the social dilemma (see Fig A in S1 Text). This shows the stronger a dilemma, a stronger set of cooperation supporting moral norms emerge.

The emergence of a set of cooperation supporting moral norms can also lead to an anti-correlation between the strategies of the individuals in the two rounds: Cooperators are more likely to play a hard strategy in the second game, and defectors are more likely to play a soft strategy in the second game. This can be seen in Fig 7C, where the connected correlation of the strategies of the individuals in the two games, $\langle s_A s_B \rangle_c = \langle s_A s_B \rangle - \langle s_A \rangle \langle s_B \rangle$, is plotted.

Here, $\langle . \rangle$ denotes an average over the population. To calculate the correlation function, I have assigned a value $+1$ to cooperation and the soft strategy, and $-1$ to the defection and hard strategies.

**Evolution of moral norms in the reputation-based model.** A similar phenomenon is at work in the reputation-based model where individuals play their two games with different opponents. In Fig 8, I turn to the reputation-based model. In Fig 8A, I plot the density of cooperators in the Prisoner's Dilemma, and in Fig 8B, I plot the density of the soft strategy in the game $B$, as a function of the probability of error in inferring the PD-strategy of the opponent, $\eta$. Here, as before, from top to bottom, game $B$ is, respectively, SD, BS, and the Leader game. Lines represent the results of the replicator-mutator dynamics, and markers show the results of simulations in a finite population. For a small probability of error, $\eta$, the model is bistable. The unbiased fixed point is plotted by a solid blue line, and the biased fixed point resulting from a biased initial condition is plotted by a dashed red line. However, for a large probability of error, the model becomes mono-stable, and the dynamics settle in a defective fixed point in which the density of strategies in both games is close to their Nash equilibrium value. I note that, for $\eta = 0.5$, individuals have no information about the strategy of their opponent. As $\eta$ increases beyond 0.5, individuals are more likely to infer the PD strategy of their opponent erroneously than by chance. This gives net information that can be used by the population to

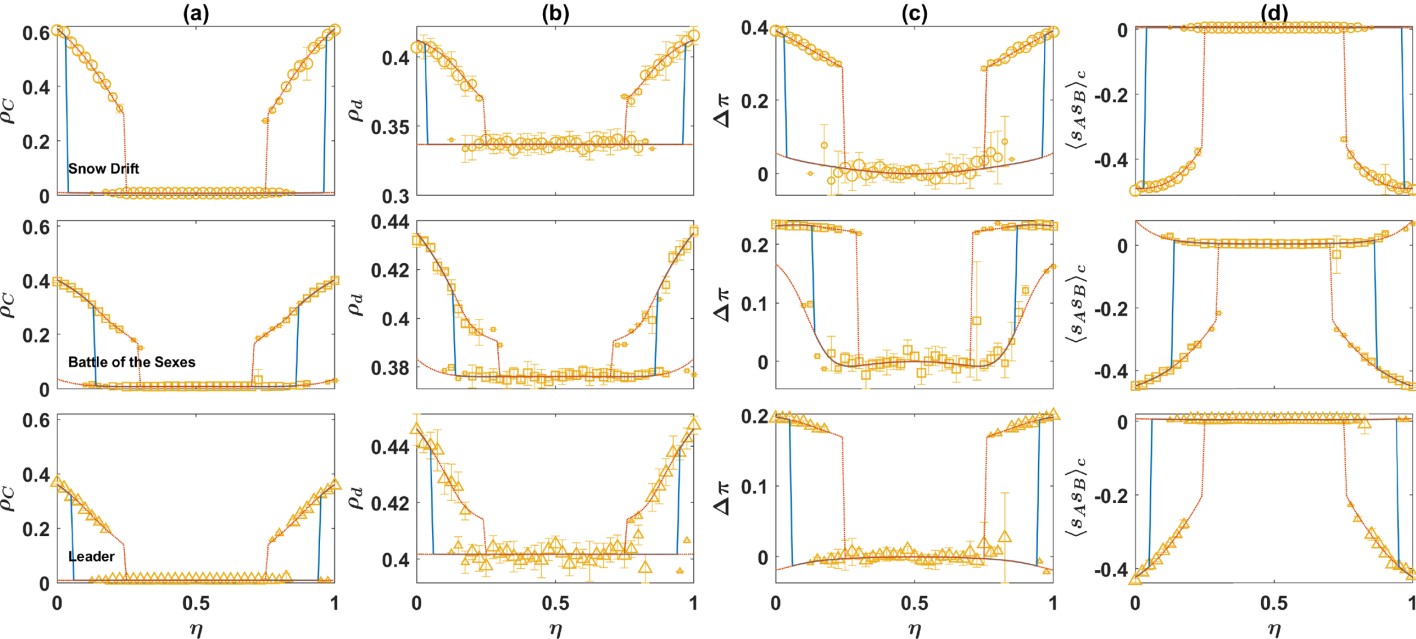

**Fig 8. The reputation-based model with the three archetypal games.** The density of cooperators, $\rho_C$, A, the density of soft strategies in game $B$, $\rho_d$, B, the normalized payoff difference of cooperators and defectors in game $B$, $\Delta\pi$, B, and the correlation between the strategy of the individuals in the two games, $\langle s_A s_B \rangle_c$, D, as a function of the probability of error in inferring the PD strategy of the opponent, $\eta$. Here, from top to bottom, the game $B$ is the Snow Drift, the Battle of the Sexes, and the Leader game. The payoff values used for the games are presented in Table 1. The lines show the result of the replicator-mutator dynamics, and the markers show the results of simulations. The solid blue line shows the equilibrium fixed point, which occurs starting from an unbiased initial condition in which the density of all the strategies are equal, and the dashed red line shows the non-equilibrium fixed point, which can occur for certain initial conditions. The system is bistable for small recognition noise, $\eta$ and both a cooperative fixed point with a high level of cooperation A and soft strategies B, and a defective fixed point with a low level of cooperation and soft strategies are possible. In the cooperative phase, cooperators receive a higher payoff from game $B$ C. Moreover, the strategies of individuals show an anti-correlation in the cooperative fixed point D, resulting from the fact that defectors play softly with cooperators and cooperators play hard with defectors in this fixed point. For the simulations, a sample of 80 simulations, in a population of size $N = 10000$ is used. The simulations start from random initial conditions. In each simulation, the dynamics settle in one of the two fixed points. The markers show the averages, and the error bars show the standard deviation in the sample of simulations that settle in the given fixed point. The size of the markers is proportional to the number of times that the given fixed point occurs. Here, the mutation rate, $\nu = 0.005$. The simulations are run for 20000 time steps, and an average over the last 1000 time steps is taken.

self-organize in a cooperative fixed point. For this reason, the dynamic is symmetric around $\eta = 0.5$ and result in the same cooperation level for $\eta$ and $1 - \eta$. Based on these considerations, in the following, I refer to the case of $\eta = 0.5$ as the maximal noise level.

For the simulations, a sample of 80 simulations, run for 20000 time steps, is used. Starting from a random initial condition, the dynamics settle in one of the stationary states. The size of the markers is proportional to the number of times that a given equilibrium occurs. As it is clear in the figure, finite-size effects such as population noise resulting in the deviation of the frequency of the strategies from the center initial condition used in solving the replicator dynamics, favor cooperation. This can be seen by noting that, in simulations in a finite population, the dynamics settle into the cooperative fixed point with a high probability, even when this is the biased fixed point in the infinite size system.

The normalized payoff difference of cooperators and defectors in the game $B$, $\Delta \pi = (\pi_C^B - \pi_D^B)/(\pi_C^B + \pi_D^B)$, is plotted in Fig 8C. As can be seen, in the cooperative fixed point, cooperators receive a higher payoff in the game $B$. As in the direct interaction model, this is due to the emergence of a set of cooperation supporting moral norms, according to which individuals are more likely to play the soft strategy with cooperators compared to defectors. This, in turn, allows cooperators, to be more likely to play a hard strategy in the game $B$. This leads to an anti-correlation between the strategy of the individuals in the two games, as can be seen in Fig 8D. On the other hand, in the defective fixed point, the payoff difference of cooperators and defectors is near zero, and almost no correlation between the strategy of the individuals in the PD and game $B$ is observed.

**Continuous variations of the structure of game $B$ and the evolution of moral norms through a symmetry breaking phase transition.** So far, using three archetypal games, we have seen cooperation and a set of cooperation supporting moral rules evolve in both models. To see how the models behave with respective to continuous variations of the structure of game $B$, I set $S = 0$, $P = 1$, $R = 3$, and $T = 5$ for the PD, and $R_B = 3$, and $P_B = 1$ for game $B$, and color plot the density of cooperators as a function of $S_B$ and $T_B$, in Fig 9A, for the direct interaction model, and in Fig 9B for the reputation-based model. In the top panels, the results of the replicator-mutator dynamics are shown, and in the bottom panel, the results of simulations in a population of $N = 1000$ individuals are shown. Here a sample of 128 simulations, run for 10000 time steps is used. Averages are taken over the last 1000 time steps.

For $S_B < 1$, and $T_B > R_B = 3$, game $B$ is a PD. In this case, cooperation does not evolve. On the other hand, for $T_B > 3$, and $S_B > 1$, game $B$ belongs to the anti-coordination class. In principle, cooperation can evolve in this case. The boundaries of bistability, plotted by markers, show the boundary above which the system becomes bistable. Below this boundary, only one fixed point, with a low level of cooperation is possible, and above this line, a cooperative fixed point emerges as well. In the bistable region, depending on the initial conditions, the dynamics settle in one of the two fixed points. The color plots show the cooperation level starting from unbiased initial conditions. A comparison of the results of the replicator-mutator dynamics and simulations in finite population shows finite size effects strongly favor cooperation, such that the transition between the two phases occurs for smaller values of $T_B$ or $S_B$ in a finite size population.

I note that the boundary of bistability is composed of two branches. Above each branch, a different fixed point occurs. Above the branch plotted by green squares, the soft strategy is $d$ (for large $T_B$), and above the branch plotted by red circles, the soft strategy is $u$ (for large $S_B$). The two branches meet at a single critical point, where the transition becomes a symmetry breaking continuous transition. Below the critical point, cooperation and defection in the PD are symmetric and are treated in the same way in game $B$. This leads to a symmetric state where the evolution of cooperation is prevented due to the cost of cooperation. Above the

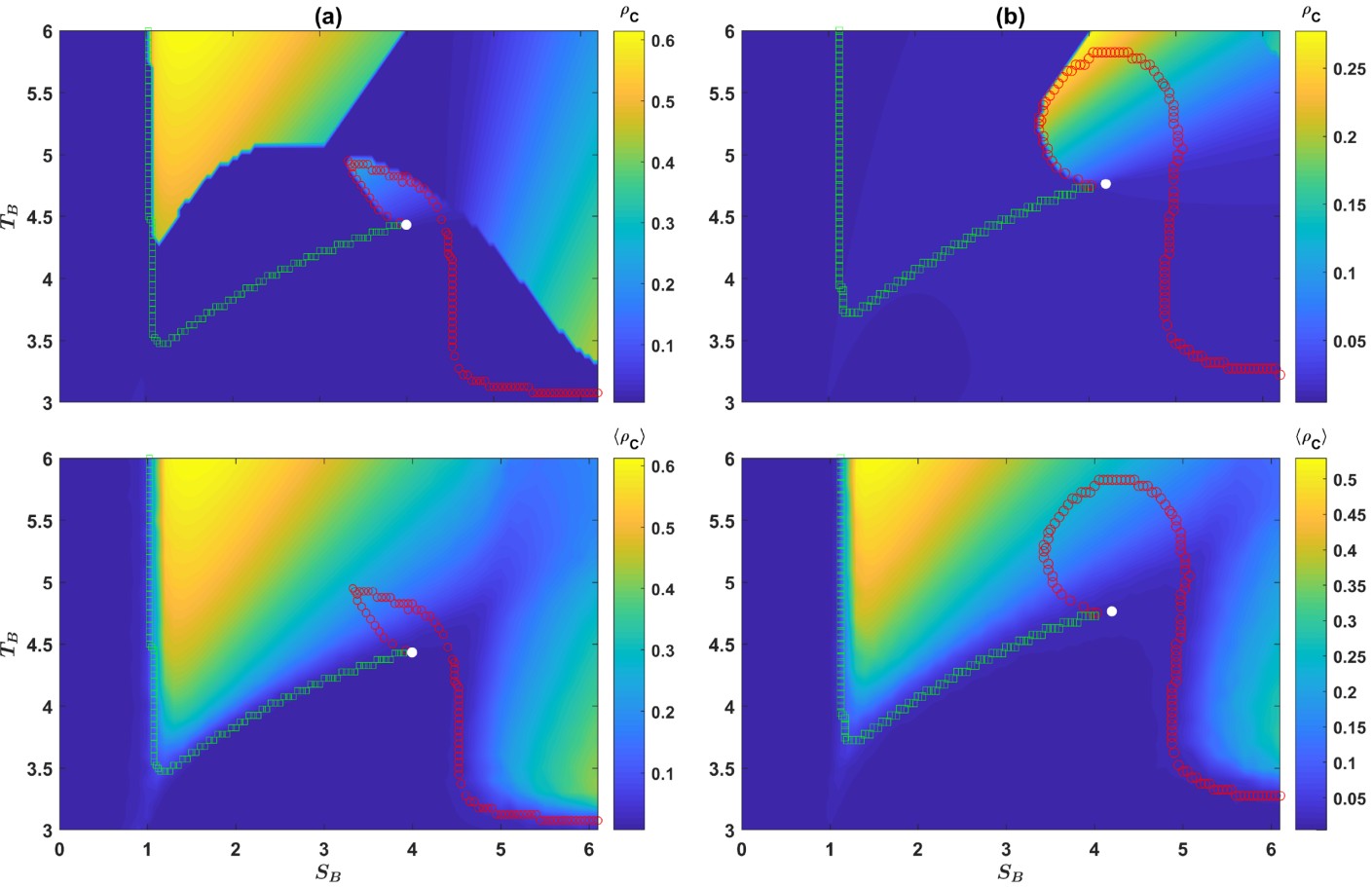

**Fig 9. The behavior of the models under continuous variation of the structure of game B.** The color plot of the density of cooperators, $\rho_C$, in the direct interaction model A, and the reputation-based model B, in the $S_B - T_B$ plane. The top panels show the result of the replicator-mutator dynamics and the bottom panels show the results of simulations in a population of 1000 individuals. In both cases, an unbiased initial condition (random assignment of strategies) is used. I have set $R = 3$, $S = 0$, $P = 1$, $T = 5$, $R_B = 3$, and $P_B = 1$. The boundaries of bistability are plotted as well. Below this boundary the dynamic is monostable, settling into a fixed point with a low level of cooperation. Above the boundary, a cooperative fixed point becomes stable and the dynamics become bistable. The two branches of the boundary meet at a critical point, where the transition becomes continuous. A comparison shows finite size effects strongly favor cooperation. Here, $\eta = 0.1$, and $v = 0.005$.

critical point, however, the symmetry between cooperative and defective strategies breaks, and a set of cooperation favoring norms emerges.

## Structured population

An interesting question is whether cooperation favoring moral norms evolves in a structured population as well? As we will see in this section, this is the case when game *B* is an anti-coordination game, but the evolution of cooperation was not observed when game *B* is a coordination game. To see why when a social dilemma is coupled with a coordination game, cooperation favoring coordination norms does not evolve in a structured population, in Fig 10A to 10C, I present snapshots of the population configuration when the game *B* is a Stag Hunt game, and in the direct interaction model. The reputation-based model shows a similar behavior. The densities of different strategies are presented in Fig 10D and 10E. The initial population is a uniformly distributed mixture of *Cdd* and *Ddu*. This initial condition favors the evolution of cooperation favoring coordination norms in a mixed population (fixed point *II* of the replicator-mutator dynamics when game *B* is a Stag Hunt game in Figs 3 and 5).

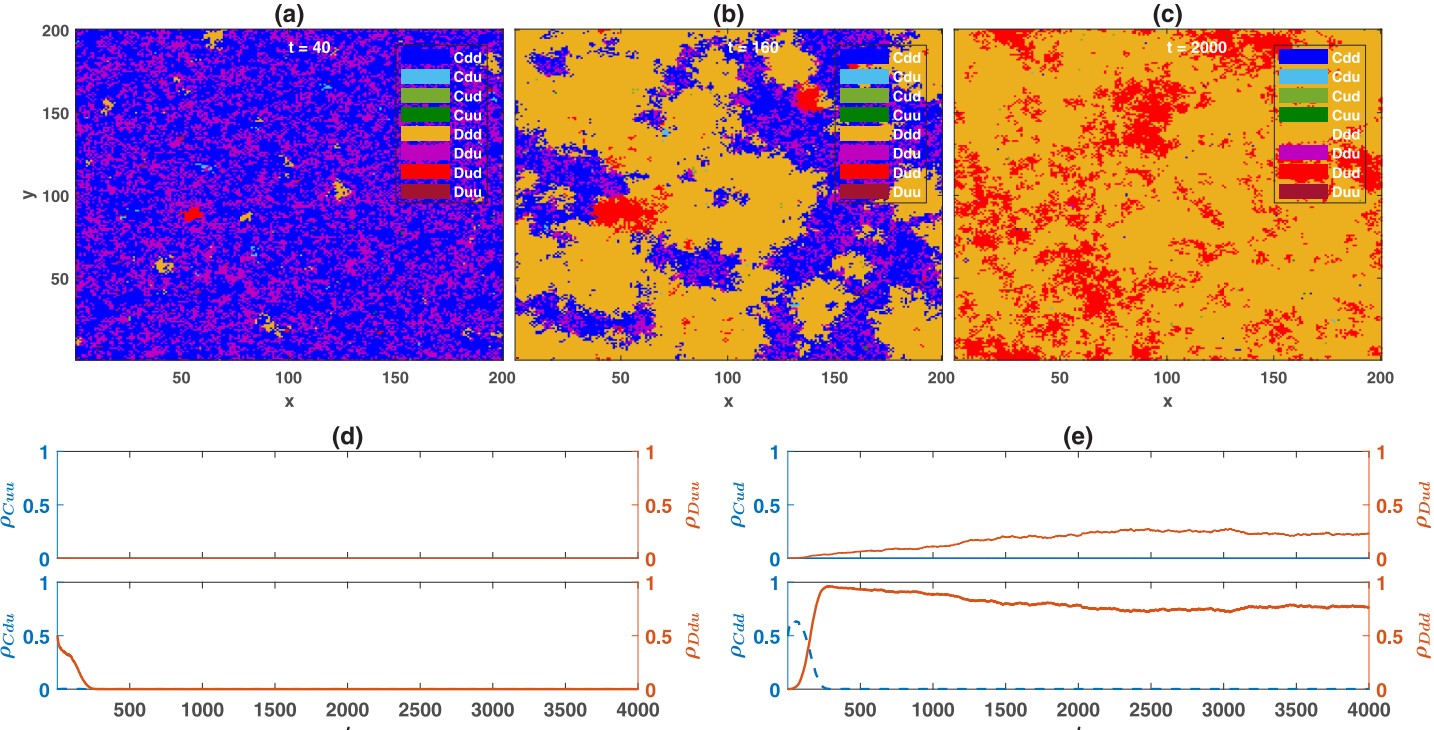

**Fig 10. Time evolution of the system in a structured population when game *B* is the Stag Hunt game.** A to C: snapshots of the population during the evolution for different times, *t*, are presented. (d) and (e): The densities of different strategies as a function of time. Here, $v = 10^{-3}$, and game *B* is the Stag Hunt game. The payoff values of the games are presented in Table 1. The initial population is a uniformly distributed mixture of *Cdd* and *Ddu*. This initial condition favors the evolution of cooperation-favoring coordination norms in a mixed population. However, due to the formation of homogeneous blocks in a structured population, cooperation favoring coordination norms are unstable, and the system evolves to a defective state composed of *Ddd* and *Dud* types. The population resides on a 200 × 200 first nearest neighbor square lattice with von Neumann connectivity and periodic boundaries. The direct interaction model is used.

However, cooperation favoring coordination norms becomes unstable in a structured population. The reason is that in a structured population, starting from any initial condition, small homogeneous blocks of similar strategies are formed. Within such homogeneous blocks, any of the consistent coordination norms (cooperators who play *d* with cooperators and defectors who play *d* with defectors) receive a high payoff and can grow. In the boundaries, defectors receive a higher payoff from the PD compared to cooperators and can replace the domain of cooperators. Consequently, the system evolves into a state where *Ddd* and *Dud* types, which defect and play *d* with defectors, dominate the population. In contrast, as we will see below, cooperation favoring moral norms evolves when game *B* is an anti-coordination game. Furthermore, importantly, population structure removes the bistability of the system and ensures the evolution of moral norms starting from all the initial conditions.

In Fig 11A I plot the density of cooperators in the reputation-based model as a function of *η*, when game *B* is one of the three archetypal games, all belonging to the anti-coordination class. This figure shows the results of simulations on a population of *N* = 40000 individuals residing on a 200 × 200 two-dimensional square lattice with the first nearest neighbor von Neumann connectivity and periodic boundaries. Cooperation in the Prisoner's Dilemma evolves as long as noise in recognition is small enough. As in the case of the mixed population, this is due to the evolution of cooperation supporting moral norms, which guarantee a higher payoff for cooperators in the game *B*. This can be seen to be the case in Fig 11B, where the normalized payoff difference of cooperators and defectors in the game *B* is plotted. Furthermore,

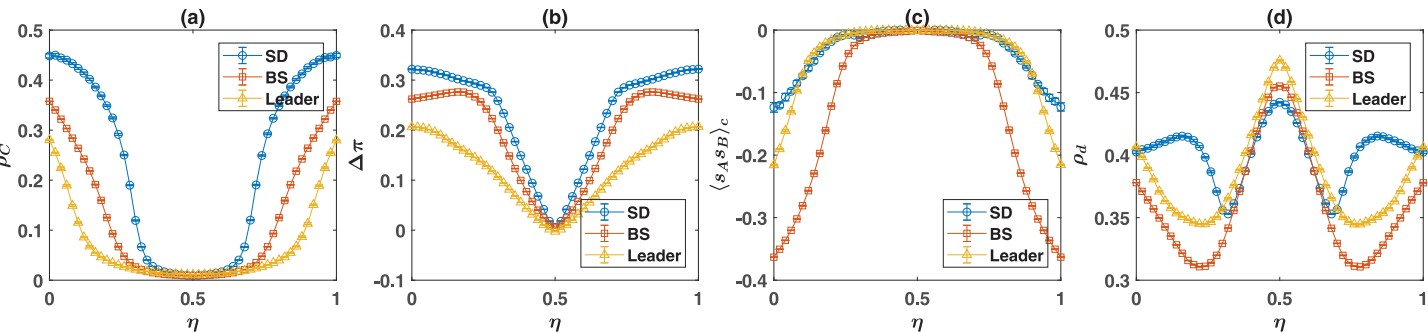

**Fig 11. The reputation-based model with three archetypal games in a structured population.** The density of cooperators, $\rho_C$, A, the normalized payoff difference of cooperators and defectors in game B, $\delta\pi$, B, the correlation between the individuals' strategies in the two games, $\langle s_A s_B \rangle_c$, C, and the density of soft strategies in game B, $\rho_d$, D, as a function of the probability of error in inferring the PD strategy of the opponent, $\eta$, are plotted. The payoff values used for the games are presented in Table 1. Simulations are performed in a population of 40000 individuals residing on a $200 \times 200$ square lattice with first nearest neighbor von Neumann connectivity and periodic boundaries. The simulations are performed for 6000 time steps, and averages and standard deviations are calculated based on the last 4000 time steps. The simulations start from random initial conditions. Here, $\nu = 0.005$.

due to the evolution of cooperation supporting norms, cooperators are more likely to play a hard strategy in the game B, than the defectors are. This leads to the negativity of the connected correlation function of the individuals' strategies in the two games, as shown in Fig 11C.

The moral system also supports a high level of soft strategies in the game B. This can be seen in Fig 11D, where the density of soft strategies in the population is plotted. The density of soft strategies is always larger than the Nash equilibrium value. This is particularly surprising, given that, although beneficial for the evolution of cooperation in the Prisoner's Dilemma, network structure can hinder the evolution of cooperation in the Snow Drift game [51]. Our results show that in contrast to what is the case in a simple strategic setting, strategic complexity can provide an avenue for network structure to play a constructive role in the evolution of cooperative behavior in the Snow Drift game.

To take a more in-depth look into the dynamics of the system, in Fig 12, I plot the density of different strategies in the population for the cases that the game B is one of the three archetypal games. As the probability of error approaches 0.5, soft strategies decrease in density. However, the density of defectors who in practice are more likely to defer to cooperators than not, that is $Ddu$ for $\eta < 0.5$, and $Dud$ for $\eta > 0.5$, increases when the noise level approaches the maximal value of 0.5. This observation shows that a higher noise level can strengthen the moral system, such that cooperation supporting norms become stronger for higher noise levels. This can be more clearly seen to be the case in Fig 13, where the densities of the strategies which play up with cooperators, $u(C)$, down with cooperators $d(C)$, up with defectors $u(D)$, and down with defectors, $d(D)$, as a function of the probability of error are plotted. As can be seen, by increasing the error probability, well close to the maximum noise level ($\eta = 0.5$), the densities of strategies that play softly with cooperators and those which play hard with defectors increase. While the density of those who play hard with cooperators and soft with defectors decreases. However, this increase in cooperation favoring strategies is not strong enough to compensate for the loss of cooperators' payoff in the game B due to increasing noise in recognition. Consequently, the density of cooperators decreases when noise approaches the maximal value.

For the maximum noise level, cooperation supporting strategies experience a rapid decline (Fig 13), and cooperation in the Prisoner's Dilemma reaches its lowest level (Fig 11A). At this point, $Dud$ and $Ddu$ become indiscriminate, and both dominate in the population. Interestingly, the density of soft strategies in the game B reaches its maximum at this point. In the case

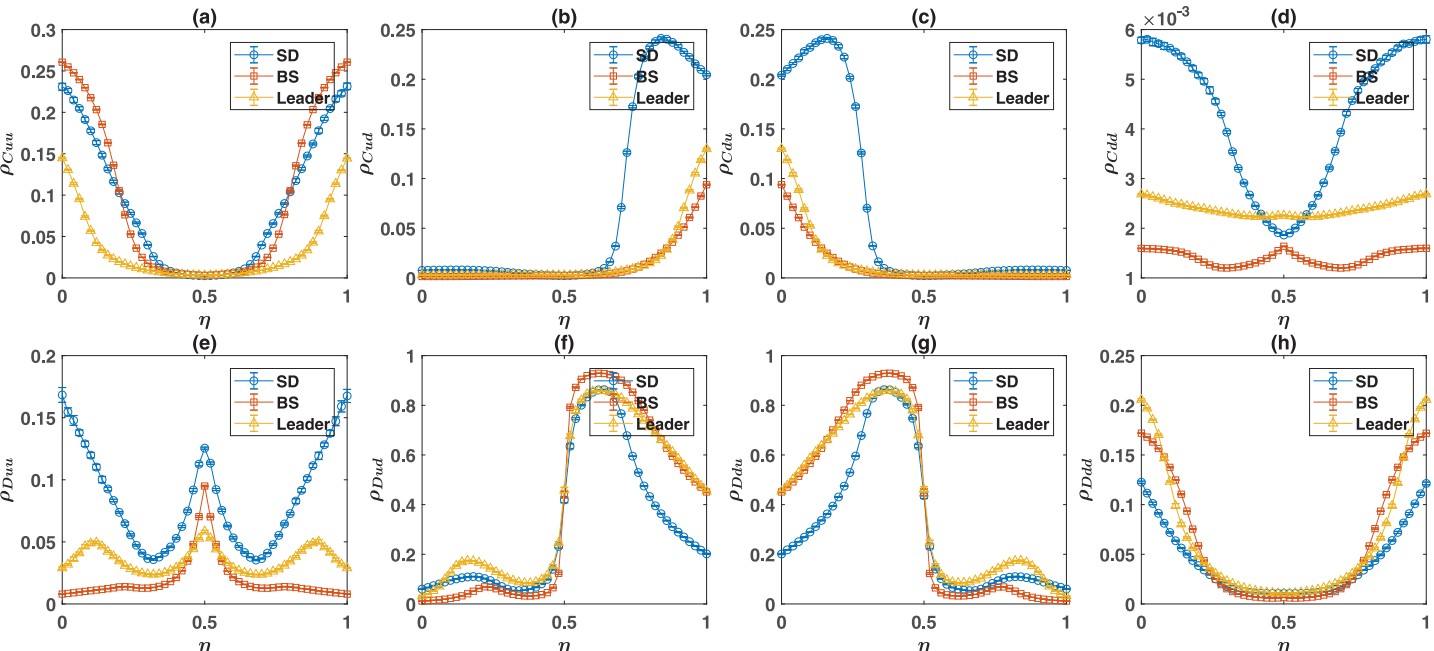

**Fig 12. The density of different strategies in the reputation-based model with three archetypal games in a structured population.** The time average density of different strategies, as a function of the probability of error in inferring the PD strategy of the opponent, $\eta$, are plotted. The payoff values used for the games are presented in Table 1. Simulations are performed in a population of size 40000 individuals residing on a $200 \times 200$ square lattice with first nearest neighbor von Neumann connectivity and periodic boundaries. The simulations are performed for 6000 time steps, and averages and standard deviations are calculated based on the last 4000 time steps. The simulations start from random initial conditions. Here, $\nu = 0.005$.

of the Snow Drift game, this value is well above the Nash equilibrium, which occurs in a mixed population. Due to the detrimental effect of network structure for the evolution of cooperation in the Snow Drift game, the density of soft strategies is even less than the Nash equilibrium value in a simple strategic setting in structured populations [51]. This observation shows that recognition noise can facilitate cooperation in the Snow Drift game in structured populations. I note that the mechanism behind this phenomenon seems to be rather independent of the

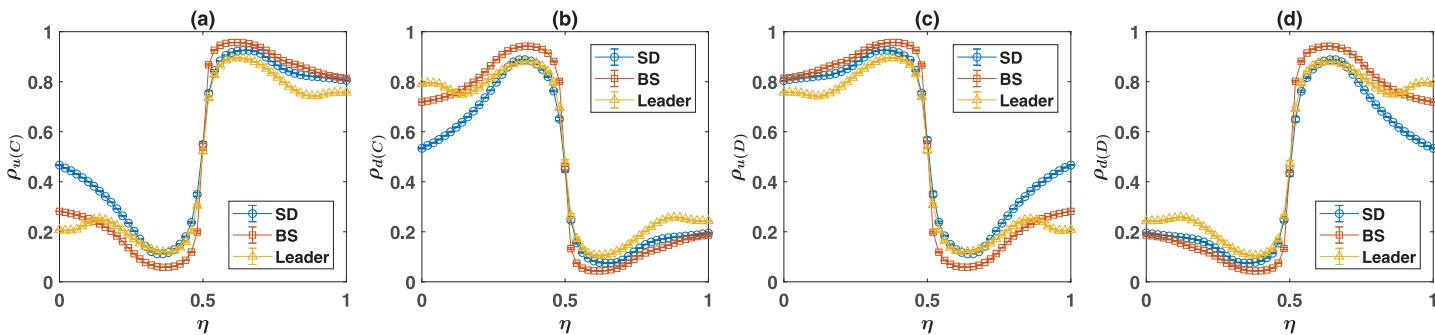

**Fig 13. Strategic response to cooperators and defectors in game B, in the reputation-based model with three archetypal games in the structured population.** The density of strategies who play up with cooperators $u(C)$, down with cooperators $d(C)$, up with defectors, $u(D)$, and down with defectors, $d(D)$, as a function of the probability of error in inferring the PD strategy of the opponent, $\eta$, are plotted. The payoff values used for the games are presented in Table 1. Simulations are performed in a population of size 40000 individuals residing on a $200 \times 200$ square lattice with first nearest neighbor von Neumann connectivity and periodic boundaries. The simulations are performed for 6000 time steps, and averages and standard deviations are calculated based on the last 4000 time steps. The simulations start from random initial conditions. Here, $\nu = 0.005$.

evolution of cooperation supporting norms, as in this case, such norms do not evolve in the system.

As the probability of error increases beyond 0.5, the density of strategies that play up with defectors and down with cooperators rapidly increases (Fig 13). Combined with the fact that in this regime, individuals are more likely to make an error in recognition of cooperators and defectors than making a correct inference, this guarantees that cooperators to be more likely to be played soft with compared to defectors. Consequently, as was the case in the mixed population, the population self-organizes into a regime where a set of cooperation supporting moral norms emerges and supports cooperation in the system.

I note that while in the mixed population, in the cooperative fixed point, *Ddu* and *Ddd* types are the only defective types which are found in large densities, in a structured population, depending on the parameter values, it can happen that all the defective strategies exist in large densities. This is due to the fact that in a structured population, domains of similar strategies are formed. While a given strategy may perform poorly globally, when surrounded by certain types, it can survive. This phenomenon, in turn, removes the bistability of the dynamics: The fate of the dynamics does not depend on the initial condition. To more closely see how this is the case, in Fig 14A to 14C, I present snapshots of the time evolution of the system, starting from a defection favoring initial condition, in which all the individuals are defectors, and a defection favoring norm prevails. That is, all the individuals are of the *Dud* type: They defect in the PD, play hard with cooperators, and soft with defectors. The time evolution of the densities of different strategies is presented in Fig 14D and 14E.

As *Dud* individuals defer to defectors, the two defective types who play hard with defectors, *Duu* and *Ddu*, reach a higher payoff. Consequently, in the first stage of the time evolution of the system, domains of *Duu* and *Ddu* form and rapidly grow in the sea of *Dud*s. However, *Dud*

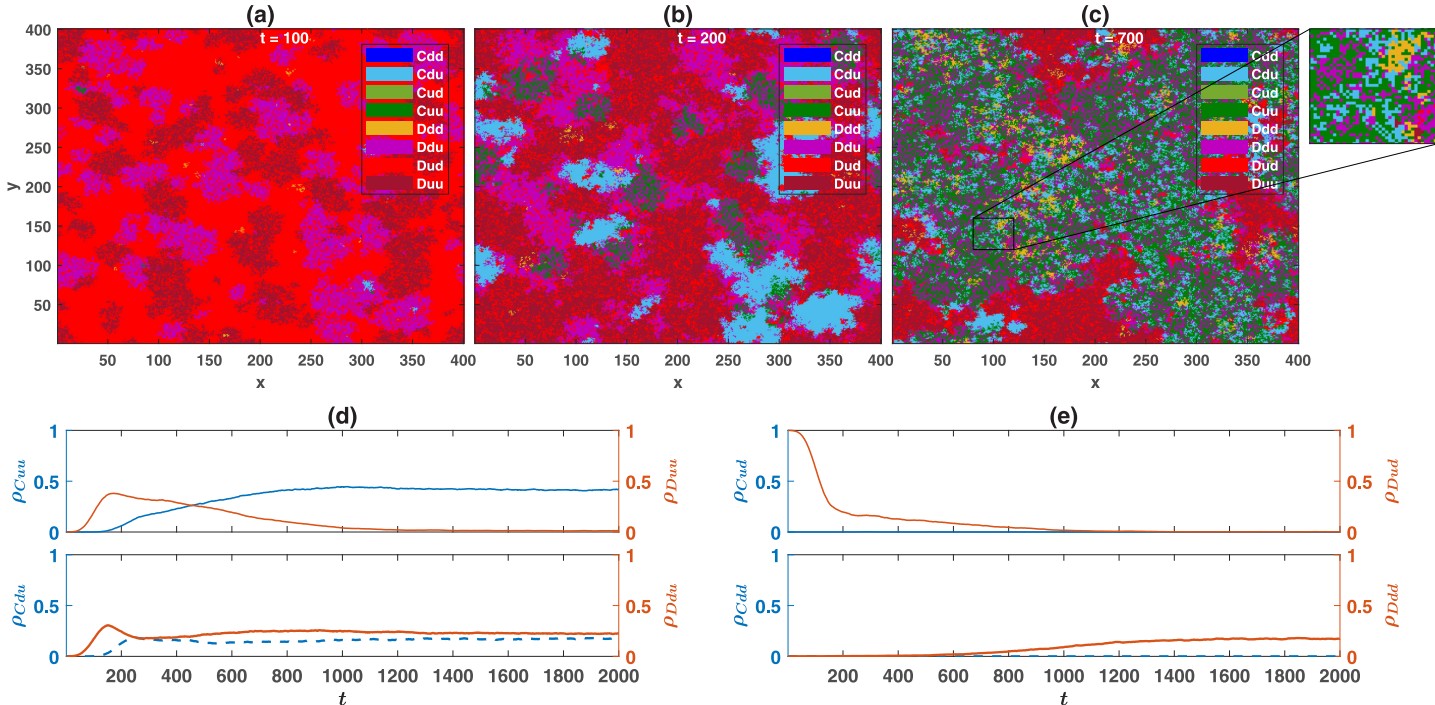

**Fig 14. Time evolution of the system.** A to C: snapshots of the population during the evolution for different times, *t*, are presented. D and E: The densities of different strategies as a function of time. Here, $v = 10^{-4}$, and game *B* is the Snow Drift game. The payoff values of the games are presented in Table 1. The initial population is of type *Dud*. The population resides on a $400 \times 400$ first nearest neighbor square lattice with von Neumann connectivity and periodic boundaries.

can coexist with *Ddu* and *Duu*. This is so because two neighboring *Ddu*s, two neighboring *Duu*s, or a *Ddu* in the neighborhood of a *Duu*, play mutually hard. On the other hand, a *Dud* by deferring to the two former strategies can perform better in their immediate neighborhood. While in a disadvantage in the sea of *Dud*s, as the *Ddu* type defers to cooperators, cooperators of type *Cdu* and *Cuu*, by playing hard against *Ddu*, reach a high payoff which compensates for the cost of cooperation, and thus, *Cdu*s and *Cuu*s grow in the *Ddu* domains. This sets the second stage of the system's time evolution, where the density of both *Cdu* and *Cuu* increases, and the density of *Ddu*, and less rapidly, *Duu* decreases. There is an important difference in the spatial patterns of the domains of *Cdu* and *Cuu* types: While *Cdu* strategies, by playing soft with each other can occupy neighboring positions, as neighboring *Cuu* types play mutually hard, they decrease the payoff of their neighbors of similar type. Consequently, *Cuu* types tend to avoid being neighbors. Instead, they form cross-like patterns where they coexist with *Ddu* type. This is crucial for the fate of the dynamics: *Cuu* and *Ddu* form a winning coalition. *Cuu* benefits *Ddu* in the PD, and *Ddu* benefits *Cuu* in the game *B*. This winning coalition can overcome the coexisting domains of *Duu* and *Dud* and invade their territories. Consequently, in the third stage of the evolution, domains of coexisting *Cuu* and *Ddu* grow by invading the domains of coexisting *Duu* and *Dud*. This is the stage where a cooperative norm slowly replaces a defection favoring norm. The densities of both *Duu* and *Dud* decreases, while the density of *Cuu* slowly increases in this stage.

Although, due to reaching a higher payoff in the game *B*, *Cuu* wins in a direct competition with *Cdu*, *Cuu*s benefit *Cdu*s in an indirect way. This is so because, due to the fact that *Cuu*s repel each other, *Cdu* can survive in the domains of coexisting *Ddu* and *Cuu*. Consequently, elimination of *Duu* and *Dud*, by the coalition of *Cuu* and *Ddu* benefits *Cdu* as well by increasing its territorial domain. For this reason, the density of *Cdu* slightly increases in the expansion phase of the *Cuu* and *Ddu* coalition. Finally, while *Ddd* performs poorly in the initial stages of the time evolution of the system, its density increases once a cooperative norm is established and *Dud* is removed. The reason is, in the presence of the anti-cooperative *Dud* type, *Ddd* performs poorly in indirect competition with both *Ddu* and *Duu*, who can exploit *Dud* better. However, once *Dud* is removed, *Ddd* can emerge in the system as well.

## Discussion

We have studied the evolution of strategies in a complex strategic setting, where individuals in a population play different games and base their strategy in a game on what happens in another game. By considering a situation where two interacting games exist, a Prisoner's Dilemma followed by a second game, we have seen that as long as the second game belongs to the coordination or anti-coordination class, the Nash equilibrium of the two-stage game can be of two types: defective equilibria, where the Nash equilibria of each of the composing games are played, and cooperative equilibria, in which coupling between games emerge and give rise to a new class of Nash equilibria not reducible to the Nash equilibria of the composing games. For the cooperative equilibrium to exist, the cost of cooperation should be smaller than a value determined by the structure of the second game. A similar condition ensures the existence of a cooperative fixed point in the evolutionary dynamics. Investigation of the evolutionary dynamics shows that a cooperative fixed point exists in a mixed population for both coordination and anti-coordination games. However, while in the former case, the cooperative fixed point has a small basin of attraction and disappears in a structured population, in the latter, a cooperative fixed point in which a set of cooperation supporting moral norms emerges and supports cooperation can evolve starting from a rather broad range of initial conditions. The evolution of moral norms, in this case, originates from the fact that it is in the individuals' best

interest to take the information about what happens in the social dilemma into account when making strategic choices in a second game to better anti-coordinate in the second game. Consequently, in the course of evolution, a set of cooperation supporting moral norms emerges based on the individual's self-interest. This appears to provide a possible mechanism for the evolution of morality in a biological population composed of self-interested individuals with simple cognitive abilities.

Importantly, population structure facilitates the evolution of a moral system by removing the bistability of the system and ensuring a cooperative state to flourish starting from all the initial conditions when the second game is an anti-coordination game. Analysis of the model in a structured population shows that noise in recognition can be beneficial for the evolution of a moral system in structured populations. Nevertheless, recognition noise also limits cooperators' ability to benefit from stronger moral norms, and thus, adversely affects cooperation. Finally, our analysis reveals in a complex strategic setting, very high levels of recognition noise facilitate the evolution of cooperative behavior in the Snow Drift game in structured populations. This shows, in contrast to what is the case in a simple strategic setting [51], network structure can be beneficial for the evolution of cooperation in the Snow Drift game. Furthermore, this provides another case for the surprisingly beneficial role that noise may play for biological functions [52–55].

Our findings provide new insights into the evolution of indirect reciprocity. By considering a simple strategic setting, namely one in which individuals only can play a social dilemma, models of indirect reciprocity have shown that specific moral rules can support an evolutionary stable cooperative state. However, the simplicity of the strategic setting requires the moral assessment module and action module to occur in the same context, which is typically a social dilemma game that individuals play. This self-referential structure can destabilize the dynamics. To solve this problem, the theory often appeal to higher-order and complex moral assessment rules. In addition to the lack of a natural mechanism to break the chain of higher-order rules, this requires a relatively high cognitive ability and a large amount of information about the past actions of the individuals in moral assessment [19], which appears to limit its applicability [20]. Furthermore, the dichotomy of moral assessment module and action module, commonly incorporated in many models of indirect reciprocity, can give rise to severe problems, when instead of public information, individuals have private information about the reputation of others [7, 59–62]. In this case, a punishment dilemma can arise: individuals may have different beliefs about the reputation of others, and thus, disagree as to what is a justified punishment [5, 59].

In contrast, as I have shown, the introduction of interaction between games circumvents these problems and leads to a simple dynamical mechanism for the evolution of a set of cooperation supporting moral norms. In this regard, in a strategically complex context, it is not necessary to define good and bad a priori to the dynamics of the system. This avoids the punishment dilemma when information is private. Nor it is necessary to define different moral assessment rules [12–14, 17, 18] and search for efficient ones [17–19, 21, 63]. Rather, the dynamics self-organizes into a symmetry broken cooperative phase where the symmetry between cooperation and defection breaks. A set of cooperation supporting moral norms evolves and costly cooperation emerges as a morally valuable or "good" trait due to a purely dynamical phenomenon and as a result of a symmetry-breaking phase transition.

As our analysis shows, a moral system not only promotes cooperation in a social dilemma, but it also increases soft strategies which can be considered as a more rational form of cooperative behavior in a second game, a strategic setting which may be a social dilemma (as in the case of SD) or may not be a social dilemma. By considering three archetypal games, and continuous variations of the structure of the second game, I have shown this is the case for a broad

range of strategic settings. In this sense, a moral system not only works to promote cooperation, but it also helps to solve coordination problems and help an efficient allocation of roles and resources. This finding seems to conform to many stylized facts about moral systems. For instance, while some moral values encourage self-sacrificing and other-regarding behavior [7, 46, 64], many other aspects of moral systems do not seem to go against individuals' self-interest, but encourage mutually beneficial behaviors, such as mutualistic cooperation [46–50], or conflict resolution [47, 49]. Fairness, loyalty, courage, respecting others, cherishing friendship, working together, and deferring to superiors are examples of such mutualistic moral values. Based on these observations, it is suggested promoting mutually beneficial behavior can provide yet another explanation for the evolution of morality [48–50]. Importantly, in our model, this second role is what makes a moral system evolvable based solely on the individuals' self-interest. In other words, the positive role of a moral system in bringing order and organization is beneficial at both the individual and group levels. This makes adherence to a moral system beneficial on an individual level and helps its evolution in a simple dynamical way. Interestingly, this aspect of a moral system acts like a Trojan Horse: Once established due to its organizing role, it also suppresses anti-social behavior and promotes cooperation and self-sacrifice.

Another aspect of the theory developed here is that the cost of cooperation can be considered as a cost paid by individuals to reach a high moral status to benefit from favorable encounters in interactions that do not involve a strict social dilemma. Interestingly, the model shows that the stronger the social dilemma and the higher the cost of cooperation, a stronger set of cooperation supporting moral norms emerges, as the likelihood that cooperators receive a favorable encounter in the game *B* increases with increasing the strength of the social dilemma (see Fig G in S1 Text for a mixed population, and Fig T in S1 Text for a structured population). Although this may not fully compensate for the higher cost of cooperation cooperators pay in stronger dilemmas, it partly alleviates cooperators' loss of payoff and helps the evolution of cooperation when the cost of cooperation is high.

Some empirical evidence has shown that costly cooperative traits can give cooperators an advantage over a diverse set of strategic contexts, such as coordination, partner choice, and conflict resolution [23–25, 65]. The prevalent theoretical understanding of these contexts, in the framework of costly signaling theory, is that the costly cooperative traits can function as an honest signal of quality based on which cooperators might receive more favorable interactions [24, 65]. More recently, an alternative theory has pointed out a simple dynamical phenomenon resulting from the density-dependent selection that can explain why cooperative traits can bring more favorable strategic responses [41]. This theory shows that the very fact that a trait is costly can lead to its scarcity, which in turn can lead to the evolution of favorable strategic responses, as such strategies do not impose a high cost on their bearer. A similar density-dependent selection can give rise to the evolution of cooperation in costly public goods [45], or consistent cooperative personalities in multistage public goods [66]. The models studied here show that a similar dynamical phenomenon can underlie the evolution of moral norms in complex strategic settings. This is the case because strategies that play softly with cooperators do not impose a high cost on their bearers due to the scarcity of costly cooperative traits. This can lead to the evolution of social norms that favor cooperative behavior over diverse issues such as conflict resolution and coordination. Counter-intuitively, once such cooperation-favoring norms evolve, the dynamics can get fixated in a cooperative state where the very tenet underlying the virtue of cooperative traits, their scarcity, get lost in a cooperative fest. This theory thus can provide an alternative explanation for the prevalence of cooperation-favoring norms in human societies, usually looked at through the lens of costly signaling theory.

The density-dependent selection underlying the evolution of moral norms has a surprising consequence: harmful social norms are just as likely to evolve as socially beneficial ones. A large body of empirical evidence has documented the existence of harmful social norms [67, 68]. Such harmful norms abound in cultures of honor, ranging from honor killing [67, 69], to pious and harmful cultural practices [68] and irrational and severe punishment [70, 71]. Inefficient gift-giving is suggested as another example of such bad social norms involving a collective loss [71]. While it is argued that reputation and reciprocity play an important role in the evolution of such harmful norms [67, 72], given their collective cost, the evolution and persistence of such harmful norms seem a puzzle. Our theory provides a simple explanation for the evolution of such detrimental norms: the cost of norms, not their benefit, determines their evolution. In other words, moral norms need to be costly for the individual but not necessarily beneficial for the group. This fact can give rise to bad norms which incur a collective cost. Surprisingly, such costly norms are just as effective as socially beneficial norms in promoting order and organization, and this phenomenon underlies their evolvability within the framework developed here. According to this viewpoint, the key to the puzzle of the evolution of harmful social norms does not rely on their potential advantage. Instead, their evolution is a consequence of the evolutionary process underlying the evolution of moral systems. Intuitively, costly traits provide a density-dependent mechanism for coordination and more efficient allocation of resources, and this fact underlies the evolution of norms that prescribe differing to such costly traits. This, counter-intuitively, can lead to an equilibrium state that costly traits are neither costly nor rare anymore because of the payoff they accrue due to favorable strategic responses. This framework appears to explain why many moral systems incorporate both socially beneficial and harmful but often individually costly elements. Furthermore, the positive role that our theory suggests that costly norms play in bringing order and organization appears to conform to the fact that culture of honors often originate and persist in law-less environments and play a crucial role in stabilizing societies in the absence of law-enforcement organizations.

## Methods

### The replicator-mutator dynamics

The model can be solved in terms of the discrete-time replicator-mutator equation, which reads as follows:

$$\rho_x(t+1) = \sum_y v_{x,y} \rho_y(t) \frac{\pi_y}{\bar{\pi}}. \tag{1}$$

Here, $\rho_x$ is the density of strategy $x$, $\pi_y$ is the expected payoff of strategy $y$, $\bar{\pi}$ is the mean payoff, and $v_{x,y}$ is the mutation rate from strategy $y$ to the strategy $x$. This can be written as:

$$v_{y,x} = \begin{cases} 1-v & if \quad y = x, \\ v/7 & if \quad y \neq x. \end{cases} \tag{2}$$

The payoff of an strategy can be written as follows. First I define:

$$\begin{aligned}
\rho_{C,u(C)} &= \rho_{Cuu} + \rho_{Cud}, & \rho_{C,d(C)} &= \rho_{Cdu} + \rho_{Cdd}, \\
\rho_{C,u(D)} &= \rho_{Cuu} + \rho_{Cdu}, & \rho_{C,d(D)} &= \rho_{Cud} + \rho_{Cdd}, \\
\rho_{D,u(C)} &= \rho_{Duu} + \rho_{Dud}, & \rho_{D,d(C)} &= \rho_{Ddu} + \rho_{Ddd}, \\
\rho_{D,u(D)} &= \rho_{Duu} + \rho_{Ddu}, & \rho_{D,d(D)} &= \rho_{Dud} + \rho_{Ddd}.
\end{aligned} \tag{3}$$

Here, the first letter in the indices shows the strategy in the PD, and $s(C)$ ($s(D)$), is the strategy in the second game against a cooperator (defector). That is, for example, $\rho_{C,u(C)}$ is the density of those individuals who cooperate in the PD and play the up strategy with cooperators. Besides, in the following, I use $\rho_C$ and $\rho_D$ for the total density of those individuals who, respectively, cooperate and defect in the PD. That is, $\rho_C = \rho_{Cuu} + \rho_{Cud} + \rho_{Cdu} + \rho_{Cdd}$ and $\rho_D = \rho_{Duu} + \rho_{Dud} + \rho_{Ddu} + \rho_{Ddd}$.

Given these definitions, the payoffs of different strategies, in the first model can be written as follows:

$$
\begin{aligned}
\pi_{Cuu} =&\ \rho_C R + \rho_D S + \rho_{C,u(C)} P_B + \rho_{D,u(C)} P_B + \rho_{C,d(C)} T_B + \rho_{D,d(C)} T_B, \\
\pi_{Cud} =&\ \rho_C R + \rho_D S + \rho_{C,u(C)} P_B + \rho_{D,u(C)} S_B + \rho_{C,d(C)} T_B + \rho_{D,d(C)} R_B, \\
\pi_{Cdu} =&\ \rho_C R + \rho_D S + \rho_{C,u(C)} S_B + \rho_{D,u(C)} P_B + \rho_{C,d(C)} R_B + \rho_{D,d(C)} T_B, \\
\pi_{Cdd} =&\ \rho_C R + \rho_D S + \rho_{C,u(C)} S_B + \rho_{D,u(C)} S_B + \rho_{C,d(C)} R_B + \rho_{D,d(C)} R_B, \\
\pi_{Duu} =&\ \rho_C T + \rho_D P + \rho_{C,u(D)} P_B + \rho_{D,u(D)} P_B + \rho_{C,d(D)} T_B + \rho_{D,d(D)} T_B, \\
\pi_{Dud} =&\ \rho_C T + \rho_D P + \rho_{C,u(D)} S_B + \rho_{D,u(D)} S_B + \rho_{C,d(D)} R_B + \rho_{D,d(D)} R_B, \\
\pi_{Ddu} =&\ \rho_C T + \rho_D P + \rho_{C,u(D)} P_B + \rho_{D,u(D)} P_B + \rho_{C,d(D)} T_B + \rho_{D,d(D)} T_B, \\
\pi_{Ddd} =&\ \rho_C T + \rho_D P + \rho_{C,u(D)} S_B + \rho_{D,u(D)} S_B + \rho_{C,d(D)} R_B + \rho_{D,d(D)} R_B.
\end{aligned}
\tag{4}
$$

Here the first two terms in each expression are the payoffs from the PD, and the last four terms are the payoff from game $B$. The validity of these expressions can be checked by enumerating all the possible strategies that a focal individual can play with. For example, the third term in the expression for $\pi_{Cuu}$ can be written by noting that a focal $Cuu$ player, meets an individual of type $C$, $u(C)$ with probability $\rho_{C,u(C)}$. In this interaction, the focal individual plays $u$ and the opponent plays $u$, leading to a payoff of $P_B$ for the focal individual. Using similar arguments, it is possible to drive expressions for the payoff of different strategies in the reputation-based model. See the Supporting Information Text, S. 2 for details.

### Mixed strategy Nash equilibria

**Cooperative mixed strategy Nash equilibria when game $B$ is a coordination game.** A mixed strategy is a set of probabilities, $\{x_{Cuu}, .., x_{Ddd}\}$, such that a strategy $i$ is played with probability $x_i$. The support of a mixed strategy is the set of all strategies which are played with nonzero probability. A mixed strategy Nash equilibrium is defined as a set of two mixed strategies, $(s, s')$ in which each strategy is the best response to the other strategy: $s = BR(s')$ and $s' = BR(s)$ [73]. This condition is achieved for a mixed strategy $s$ if the payoff of all the strategies in the support of a mixed strategy is the same, and no other strategy outside of the support gives a higher payoff against the mixed strategy [73]. The first criteria can be satisfied by solving a set of linear equations to achieve indifference of each player over the support of their mixed strategy. To see this, consider the mixed strategy corresponding to the cooperative fixed point $I$ of the evolutionary dynamics when game $B$ is the Stag Hunt game. The expected payoff of the strategy $Cdd$ against a mixed strategy, $(x_{Cdd}, c_{Ddu})$ is $(R + R_B)x_{Cdd} + (S + R_B)x_{Ddu}$ and the expected payoff of $Ddu$ is equal to $(T + R_B)x_{Cdd} + (P + S_B)x_{Ddu}$. Equaling these payoffs and setting the normalization condition, $x_{Cdd} + x_{Ddu} = 1$, gives a set of two equations which can be solved to give:

$$
\begin{aligned}
x_{Cdd} &= \frac{P - R_B - S + S_B}{P + R - R_B - S + S_B - T}, \\
x_{Ddu} &= \frac{R - T}{P + R - R_B - S + S_B - T}.
\end{aligned}
\tag{5}
$$

It is also easy to check that within the general parametrization of the Stag Hunt game, no other strategy gives a higher payoff than *Cdd* and *Ddu* against this mixed strategy (the highest payoff outside of the support is reached by *Ddd* which is equal to that reached by *Ddu* and *Cdd*).

Using similar steps, it is possible to derive the mixed strategy corresponding to the fixed point *II* of the evolutionary dynamics when game *B* is a Stag Hunt game as follows:

$$
\begin{aligned}
x_{Cud} &= \frac{P + P_B - R_B - S}{P + 2P_B + R - 2R_B - S - T}, \\
x_{Ddu} &= \frac{P_B + R - R_B - T}{P + 2P_B + R - 2R_B - S - T}.
\end{aligned}
\tag{6}
$$

**Cooperative mixed strategy Nash equilibria when game *B* is an anti-coordination game.** The mixed strategy Nash equilibria corresponding to the cooperative fixed point of the evolutionary dynamics when game *B* is an anti-coordination game involves the strategies *Cuu*, *Cdu*, *Ddu*, and *Ddd*, and can be derived using similar steps. Using the helping game version of the Prisoner's dilemma, this mixed strategy is given by the following expression:

$$
\begin{aligned}
x_{Cuu} &= \frac{(R_B - T_B)(cP_B + cR_B + P_B R_B - cS_B - (c + P_B + R_B)T_B + T_B^2)}{(P_B + R_B - S_B - T_B)(S_B^2 + P_B(2R_B - S_B - T_B) + T_B^2 - R_B(S_B + T_B))}, \\
x_{Cdu} &= \frac{(P_B - S_B)(cP_B + cR_B + P_B R_B - cS_B - (c + P_B + R_B)T_B + T_B^2)}{(P_B + R_B - S_B - T_B)(S_B^2 + P_B(2R_B - S_B - T_B) + T_B^2 - R_B(S_B + T_B))}, \\
x_{Ddu} &= \frac{(T_B - R_B)[(P_B - S_B)(-R_B + S_B) + c(P_B + R_B - S_B - T_B)]}{(P_B + R_B - S_B - T_B)(S_B^2 + P_B(2R_B - S_B - T_B) + T_B^2 - R_B(S_B + T_B)))}, \\
x_{Ddd} &= \frac{(S_B - P_B)[(P_B - S_B)(-R_B + S_B) + c(P_B + R_B - S_B - T_B)]}{(P_B + R_B - S_B - T_B)(S_B^2 + P_B(2R_B - S_B - T_B) + T_B^2 - R_B(S_B + T_B))}.
\end{aligned}
\tag{7}
$$

These expressions describe the cooperative fixed point of the evolutionary dynamics for all the three games belonging to the anti-coordination class considered. Using the general formulation of the Prisoner's Dilemma, it turns out that the fixed point only depends on two combinations, $T - R$ and $P - S$, which are equal to the cost of cooperation in the Helping game version of the Prisoner's dilemma. This implies that the condition for the evolution of cooperation only depends on the parameters of the Prisoner's dilemma through the cost of cooperation.

**Fully cooperative mixed strategy Nash equilibrium.** As we have seen, when game *B* is a Snow Drift game, for small cost of cooperation, the evolutionary dynamics of the two-stage game has a fully cooperative fixed point which is composed only of the cooperative strategies. This fixed point corresponds to a fully cooperative mixed strategy Nash equilibrium. To derive this equilibrium, consider the mixed strategy defined by the probabilities, $x_{Cuu}$, $x_{Cud}$, $x_{Cdu}$, $x_{Cdd}$, and zero probability of playing the defective strategies. The expected payoff of *Cuu* and *Cud* is equal to $x_{Cuu}(R + P_B) + x_{Cud}(R + P_B) + x_{Cdu}(R + T_B) + x_{Cdd}(R + T_B)$ and the expected payoffs of *Cdu* and *Cdd* is equal to $x_{Cuu}(R + S_B) + x_{Cud}(R + S_B) + x_{Cdu}(R + R_B) + x_{Cdd}(R + R_B)$. Equating these payoffs, subject to the normalization condition $x_{Cuu} + x_{Cdu} + x_{Cud} + x_{Cdd} = 1$, we derive for the probabilities:

$$
\begin{aligned}
x_{Cuu} &= \frac{R_B - T_B}{P_B + R_B - S_B - T_B} - x_{Cud}, \\
x_{Cdu} &= \frac{P_B - S_B}{P_B + R_B - S_B - T_B} - x_{Cdd}.
\end{aligned}
\tag{8}
$$

For this strategy profile to be a Nash equilibrium, the payoff of the strategies in the support should be at least as high as the payoff of the all the strategies outside the support of the mixed strategy. The payoffs of $Duu$ and $Dud$ strategies against this mixed strategy equals $x_{Cuu}(T + S_B)$ $+ x_{Cud}(T + T_B) + x_{Cdu}(T + P_B) + x_{Cdd}(T + T_B)$, and the payoffs of $Ddu$ and $Ddd$ equals $x_{Cuu}(T + S_B) + x_{Cud}(T + R_B) + x_{Cdu}(T + S_B) + x_{Cdd}(T + R_B)$. The payoff of all the strategies in the support, which are now equalized using Eq 8, is equal to $\frac{R_B - T_B}{P_B + R_B - S_B - T_B}(R + P_B) + \frac{R_B - T_B}{P_B + R_B - S_B - T_B}(R + T_B)$. Requiring the payoffs of the defective strategies to be larger than this value,

$$x_{Cuu}(T + S_B) + x_{Cud}(T + T_B) + x_{Cdu}(T + P_B) + x_{Cdd}(T + T_B) \geq$$
$$\frac{R_B - T_B}{P_B + R_B - S_B - T_B}(R + P_B) + \frac{R_B - T_B}{P_B + R_B - S_B - T_B}(R + T_B),$$
$$x_{Cuu}(T + S_B) + x_{Cud}(T + R_B) + x_{Cdu}(T + S_B) + x_{Cdd}(T + R_B) \geq$$
$$\frac{R_B - T_B}{P_B + R_B - S_B - T_B}(R + P_B) + \frac{R_B - T_B}{P_B + R_B - S_B - T_B}(R + T_B),$$

$$(9)$$

subject to Eq 8, and using the payoff values of the Snow Drift game, we arrive at the condition:

$$x_{Cud} + x_{Cdu} < 1/3 - c/2. \tag{10}$$

Eqs 8 and 10 define a set of two equations that give infinitely many fully cooperative mixed strategy Nash equilibria. By using Eq 10, it can be seen that such fully cooperative mixed strategy Nash equilibria exist only if the cost of cooperation is smaller than 2/3, which agrees with the results from the replicator-mutator dynamics presented in Fig 4. It is possible to derive conditional expressions for the existence of a fully cooperative Nash equilibrium for a general form of the game $B$ belonging to the anti-coordination class. By using such expressions, it is possible to show that no fully cooperative mixed strategy Nash equilibrium exists when the game $B$ is the leader or the Battle of the Sexes, given by the payoff values presented in Table 1.

## Simulations and numerical solutions

Numerical solutions result from numerically solving the replicator-mutator dynamics. Simulations are performed based on the model definition. Matlab codes used in simulations and numerical solutions are given in the Supporting Information Text, S. 9. The base payoff values used in this study (unless otherwise stated) are presented in Table 1. See Supporting Information Text for more details on simulations and analytical calculation.

## Supporting information

**S1 Text. Supporting information text.** Overview of the models, details of methods and further analysis of the models.
(PDF)

**S1 Video. Supplementary Video 1.** Illustration of the dynamics of the model in a structured population for the interaction of the Prisoner's Dilemma and the snow drift game.
(AVI)

**S2 Video. Supplementary Video 2.** Illustration of the dynamics of the model in a structured population for the interaction of the Prisoner's Dilemma and the Battle of the Sexes.
(AVI)

**S3 Video. Supplementary Video 3.** Illustration of the dynamics of the model in a structured population for the interaction of the Prisoner's Dilemma and the Leader game.
(AVI)

## Author Contributions

**Conceptualization:** Mohammad Salahshour.

**Data curation:** Mohammad Salahshour.

**Formal analysis:** Mohammad Salahshour.

**Funding acquisition:** Mohammad Salahshour.

**Investigation:** Mohammad Salahshour.

**Methodology:** Mohammad Salahshour.

**Project administration:** Mohammad Salahshour.

**Resources:** Mohammad Salahshour.

**Software:** Mohammad Salahshour.

**Supervision:** Mohammad Salahshour.

**Validation:** Mohammad Salahshour.

**Visualization:** Mohammad Salahshour.

**Writing – original draft:** Mohammad Salahshour.

**Writing – review & editing:** Mohammad Salahshour.

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
