## [Decision Letter · Decision Letter 0]

2 Jun 2021

Dear Dr. Salahshour,

Thank you very much for submitting your manuscript "Playing soft with cooperators emerges as a moral norm and promotes cooperation in evolutionary games" for consideration at PLOS Computational Biology.

As with all papers reviewed by the journal, your manuscript was reviewed by members of the editorial board and by several independent reviewers. In light of the reviews (below this email), we would like to invite the resubmission of a significantly-revised version that takes into account the reviewers' comments.

The reviewers recognize the originality of the proposed model and acknowledge that the paper has potential to be a valuable contribution. At the same time, however, the reviewers raise pertinent concerns related with the 1) readability and organization of the manuscript; 2) lack of theoretical analysis and intuition for some of the results observed; 3) lack of decisive evidence that the conclusions are due to the described mechanism (i.e., asymmetry of the second game equilibrium).

While we recognize the potential of this work and the comprehensiveness of the numerical analysis performed, we share the concerns pointed by the reviewers. We believe that the manuscript should be revised to ensure that the cooperation mechanism proposed is comprehensible and the conclusions general. All points raised by the reviewers should be carefully addressed in a revised version.

- The readability and logic flow of the paper must be improved (for example, following Reviewer #1 and Reviewer #2 excellent suggestions). As Reviewer #3 advises, details on the motivation for the mechanism and the biological relevance of the model should also be substantiated.

- It is relevant to provide a more general analysis of the model presented, at least for the baseline setting studied (i.e., well-mixed and direct interactions). As Reviewer #2 recommends, one option is to perform a static analysis of the 2-stage game proposed. This could provide a more precise description of the equilibria that one expects in the coupled game, which in turn can inform the general properties of replicator equations studied. This theoretical treatment can also improve the paper along Reviewer #3 remarks, by helping to clarify whether the results presented hold for every game B with an asymmetric equilibrium.

- Finally, it should be clarified whether the results obtained are due to the asymmetry of the second game played. As Reviewer #2 inquires, will similar improvements in cooperation ensue if the second game is a Stag-Hunt? Or, as Reviewer #3 suggests, is it sufficient that the second game has a mixed strategy Nash equilibrium? Is the asymmetric equilibrium of the second game a sufficient or necessary condition for cooperation in the first game to be stable?

We cannot make any decision about publication until we have seen the revised manuscript and your response to the reviewers' comments. Your revised manuscript is also likely to be sent to reviewers for further evaluation.

Sincerely,

Fernando P. Santos

Guest Editor

PLOS Computational Biology

Ville Mustonen

Deputy Editor

PLOS Computational Biology

The paper has been reviewed by three reviewers. The reviewers recognize the originality of the proposed model and acknowledge that the paper has potential to be a valuable contribution. At the same time, however, the reviewers raise pertinent concerns related with the 1) readability and organization of the manuscript; 2) lack of theoretical analysis and intuition for some of the results observed; 3) lack of decisive evidence that the conclusions are due to the described mechanism (i.e., asymmetry of the second game equilibrium).

While we recognize the potential of this work and the comprehensiveness of the numerical analysis performed, we share the concerns pointed by the reviewers. We believe that the manuscript should be revised to ensure that the cooperation mechanism proposed is comprehensible and the conclusions general. All points raised by the reviewers should be carefully addressed in a revised version.

- The readability and logic flow of the paper must be improved (for example, following Reviewer #1 and Reviewer #2 excellent suggestions). As Reviewer #3 advises, details on the motivation for the mechanism and the biological relevance of the model should also be substantiated.

- It is relevant to provide a more general analysis of the model presented, at least for the baseline setting studied (i.e., well-mixed and direct interactions). As Reviewer #2 recommends, one option is to perform a static analysis of the 2-stage game proposed. This could provide a more precise description of the equilibria that one expects in the coupled game, which in turn can inform the general properties of replicator equations studied. This theoretical treatment can also improve the paper along Reviewer #3 remarks, by helping to clarify whether the results presented hold for every game B with an asymmetric equilibrium.

- Finally, it should be clarified whether the results obtained are due to the asymmetry of the second game played. As Reviewer #2 inquires, will similar improvements in cooperation ensue if the second game is a Stag-Hunt? Or, as Reviewer #3 suggests, is it sufficient that the second game has a mixed strategy Nash equilibrium? Is the asymmetric equilibrium of the second game a sufficient or necessary condition for cooperation in the first game to be stable?

Reviewer's Responses to Questions

**Comments to the Authors:**

Reviewer #1: This manuscript theoretically explores an emergent mechanism of cooperation that supports moral norms by using both analytical solutions using replicator dynamics and individual-based simulations. In a prisoner's dilemma game and subsequent asymmetric Nash game, they show that a kind of payoff adjustment mechanism works by deciding the strategy of the subsequent game according to the strategy of the PD game, and as a result, a cooperative regime emerge.

This manuscript has sufficient originality and detailed analysis and may reach the publication level of PlosCB. For that purpose, it is necessary to reorganize the treatise so that it is easy for the readers to understand, mainly from the viewpoint of ingenuity of presentation. There are other improvements as well.

1) I think that a phase diagram towards cooperation is summarized as follows: Rare C-> [*s*] predominates-> phase transition-> [C*h] and [Ds*] dominate. Is my understanding correct? Anyway, the additional conceptual diagrams that depict the mechanism will help audience to understand clearly.

2) On lines 243-246, they claim that an asymmetric Nash, not a PD, is a necessary condition, but there is no evidence there. The reader should wait from line 301 to line 310 while maintaining frustration. Furthermore, the discussions on lines 180-184 and 248-251 are very important, but the conclusion is placed on lines 257-277. A well-streamed presentation requires a restructuring of the logic.

3) The captions in Figures 1 and 2 and the main text are redundant. Moreover, the captions themselves are not enough to fully understand those figures. For example, I don't know that DeltaPai indicates whether C or D is dominant. There is a typo in the definition of DeltaPai on line 177.

4) You need to correct the variable mismatch. which one is correct? {d, u} (line 88, Figure 7, Figure 8 and results section) or {s, h} (Figure 3). You have to unify them. In addition, they mention "R = 128 and T = 10000" on line 286, where R and T are payoff parameters. The same variable name should be avoided.

5) When I read line 172, I feel that the author implicitly thinks "soft strategy = cooperation". Is that so?

Reviewer #2: (A) Summary

In this paper, the author explores the evolutionary dynamics that arises when players sequentially engage in two games. The first game is assumed to be a prisoner's dilemma. The second game is an anti-coordination game with asymmetric equilibria (such as the snowdrift game). Importantly, behavior in the second game may depend on the co-player's behavior in the first game (e.g., a player may reward co-players who cooperated in the first game by coordinating on their preferred equilibrium in the second game).

The author explores several variants of this model,

a) depending on whether or not the same players engage in the two games ("direct interaction" vs "reputation-based" model), and

b) depending on whether the population is well-mixed or whether games take place on a two-dimensional lattice.

The paper reports a two-fold advantage of coupling the two games:

a) the coupling makes it more likely that players cooperate in the prisoner's dilemma

b) the coupling makes it easier to coordinate on an equilibrium of the second game

(B) Overall assessment

In my opinion, the considered setup is quite interesting and understudied. The relevant question is: Even if we know evolution in each of simple normal-form games, which further effects can arise when two or more of these games are coupled in a non-trivial way. The current article gives some interesting answers. It shows that such a coupling can simultaneously enhance cooperation in a prisoner's dilemma and coordination in the snowdrift game.

In addition, I'd like to positively mention that the author has obviously put quite some effort into the analysis of this framework. The results are shown for both well-mixed and structure populations, and in a setup where players interact directly in the two games, or indirectly. In some sense, there are sufficiently many results in this paper to provide material for two papers.

Having said that, I need to admit I found it rather difficult from reading the paper to really understand which mechanism is driving the presented results. I believe there are two reasons why I found it difficult:

a) The results are entirely numerical (either solutions of the replicator equation are plotted, or simulation results).

b) Especially in the beginning of the results section, the author's pace is too quick.

As such, the results too often have a "black box" feeling -- the final results are reported but the underlying mechanisms that drive these results remain somewhat unclear. As a result, I'm not convinced that all the conclusions that the author draws are actually confirmed by the model.

Overall, I believe that the above weaknesses can be addressed, but it will require quite some additional effort. I give some more specific suggestions below. I am aware that this additional work makes the paper even longer; but I really believe this effort would help readers considerably to make sense of some of the results.

If the author is able to describe more clearly which mechanisms drive the observed results, I believe the paper is appropriate for PLoS Computational Biology.

(C) Specific comments

(C.1) The paper would greatly benefit from a clear and detailed analysis of a simple baseline case, before even going into the evolutionary analysis. For example, the author could take the case of a well-mixed population, in the direct-interaction setup, where players play the standard repeated prisoner's dilemma in the first round (with R=3, S=0, T=5, P=1) and the snowdrift game in the second round (with R=3, S=1, T=5, P=0).

The author could use this example to clearly describe the 8 possible strategies. Then the author could even display the entire 8x8 payoff matrix.

To explain the basic logic of the game, the author could then provide a static equilibrium analysis of this baseline game. For example, what is the set of all subgame perfect equilibria (or alternatively, of all Nash equilibria) of this game? For those equilibria that lead to at least some cooperation in the first game, please describe the logic why cooperation in the first game can be stable.

This static analysis would help in several ways:

a) It provides some intuition for the subsequent evolutionary analysis

b) It immediately explains the fixed points of replicator dynamics (since any Nash or subgame perfect equilibrium is a fixed point of replicator dynamics)

c) It makes the reader more familiar with the setup before attempting to understand the effect of various parameter changes (note that in the present version, already the very first figure shows the non-trivial effects of changing the temptation payoff in the prisoner's dilemma).

I am not sure how difficult it would be to offer a similar static analysis for the general case (where the first game is an arbitrary prisoner's dilemma and the second game is an arbitrary anti-coordination game). If possible, this would be a useful addition to the SI. It would help to explain how the feasibility of cooperation in the prisoner's dilemma depends on the payoff parameters. However, it might well be that such a general analysis is too difficult to accomplish, and I don't insist on it.

(C.2) Related to the previous point, the author mentions at several places that for cooperation in the prisoner's dilemma to be feasible, the second game needs to have an asymmetric equilibrium.

I'm not convinced this is true. To see why, assume that the second game is a coordination game, like Stag-Hunt, with payoffs R=5, S=0, T=1, P=1. Consider the following strategy: Cooperate in the prisoner's dilemma; then play Stag (i.e., the more efficient equilibrium) if both players cooperated; otherwise play Hare (the less efficient equilibrium). Then one can show:

a) if both players adopt this strategy, they both cooperate in the prisoner's dilemma

b) it is a subgame perfect equilibrium (and hence a fixed point of replicator dynamics).

Of course, the above strategy requires that players can condition their second-game behavior on the previous behavior of both players. However, it seems to me this example suggests that it's not the asymmetry of equilibria that is necessary for the results. Rather it is the fact that the second game needs to have more than one equilibrium (symmetric or asymmetric), and one of the equilibria is more profitable to a player than the other. I would thus like the author to discuss the case of a stag-hunt game as the second game more specifically.

(C.3) One thing that I find hard to understand is the case of T=3 in Figure 1 for the snowdrift game. In this case, all of the players seem to cooperate in the prisoner's dilemma. However, if everyone cooperates in the prisoner's dilemma, it is no longer possible to use the first-game behavior as an effective coordination device for the second game. However, if the first round behavior does not help coordination in the second game any longer, why is this equilibrium routinely selected by individual based simulations? Could the author explain this case in more detail?

(D) Minor comments

(D.1) It seems that in each case, the author only looks at one specific trajectory of replicator dynamics: the initial population is always assumed to consist in equal proportions of all players.

This may appear "fair", but as a method to analyse a dynamical system it is rather unorthodox. When analysing such systems, I'd rather try to first describe all stable and unstable fixed points. In a second step, I'd then try to numerically estimate the basin of attraction of each fixed point.

I don't think this is an important point, because the author's simulations are done for random initial populations (not just the perfect initial population that consists in equal proportion of all available strategies). However, I think it would be helpful to make it more explicit that only one single trajectory of replicator dynamics has been studied. In addition, I would avoid terminology like "equilibrium fixed point" and "non-equilibrium fixed point" (as in lines 208-209). There is nothing special about the fixed point that is reached from a uniform initial distribution of strategies.

(D.2) Related to the previous point, when simulations and the solution according to replicator dynamics disagree, the author often speaks of a "finite size effect". However, given that the population size in most simulations is pretty large, I'm not convinced it's indeed the finiteness of the population that makes the difference. Rather I could well imagine that the difference is due to the fact that simulations are really based on random initial populations, whereas the author only studies replicator dynamics for a very specific initial population.

(D.3) Line 56: At this point of the text, it is difficult to understand what it means to "play softly with cooperators", because the meaning of "soft" hasn't been explained yet. Maybe it's better to just say "play the strategy that gives a higher payoff to cooperators"

(D.4) Line 124: I believe the appropriate condition should read "T+S < 2R" instead of "T<2R" (note that this is inconsequential, because S is set to zero for this prisoner's dilemma).

(D.5) Figures 1-2: It would be very helpful if each row had a header saying "snowdrift game", "battle of sexes" and "leader", respectively. Also, it would be useful if the y-axis ranges from -0.05 to 1.05 [instead of 0 to 1], such that one can better see fixed points that are exactly on the boundary.

(D.6) In Figure 2, please say which value of T is used for this simulation (I assume T=5, but it is better to be sure). Also, I don't think the meaning of the variable nu has been explained at this point, so please write "Here, the mutation rate is nu=0.005" instead of "Here, nu=0.005".

(D.7) I don't think it is very useful to consider error-values larger than 0.5. An error-rate eta > 0.5 just means that the labels "C" and "D" switch their meaning. For example, in the extreme case, eta=1, there are basically no errors at all, just players start calling each C of the co-player "D".

Of course, the author is aware of this, and the results are explained appropriately. But still, I think the paper would become clearer if the error rate eta was restricted to be at most 0.5 from the outset (and to only show the left half of each panel in Figure 2, for example).

(D.8) In line 312-316, the author argues that in typical models of structured populations, network reciprocity can only promote cooperation if the individuals are selected for reproduction with a probability proportional to the exponential of their payoff. I don't think this statement is true (and the author does not provide any reference that shows that this exponential form is indeed necessary).

(D.9) Lines 397, 400: "see of Duds" should probably read "sea of Duds"

(D.10) It is somewhat confusing that in the SI page 2, the first strategy is called "down" and the second is called "up". Usually it's the other way round.

Reviewer #3: In this article, the author uses numerical simulations to study a model in which individuals first make a choice in a Prisoner’s dilemma and then make a choice in a game B. Reputation is formalised through a probability of guessing the choice of the partner (in the B game) in the previous PD, before playing B. It is argued that, whenever B has a non-symmetric equilibrium, cooperation may evolve in this model, both in structured and mixed populations.

I think that the result is overstated. The model is tested only in some particular games B, therefore it is impossible to conclude that the results hold for every game B with a non-symmetric equilibrium. Moreover, the paper is missing a theoretical analysis. My feeling is that, whenever B has an equilibrium in mixed strategies, then cooperation in PD might be supported in equilibrium of the sequence of games, and thus one might have the evolution of cooperation. So, my feeling is that the paper fails to convince the reader that the issue preventing the evolution of cooperation is really the symmetry of the Nash equilibrium of B, or rather is the support of the equilibria. Another limitation is the lack of motivation: what are real life examples of B? Why shall we care about this method to promote cooperation? What is the biological relevance of this model?

In sum, I do think that this paper has the potential to make a valuable contribution, but, at this stage, it is at a too early stage to be considered for publication, especially in a highly selective journal such as Plos CB.

**Have the authors made all data and (if applicable) computational code underlying the findings in their manuscript fully available?**

Reviewer #1: Yes

Reviewer #2: Yes

Reviewer #3: None

PLOS authors have the option to publish the peer review history of their article (what does this mean?). If published, this will include your full peer review and any attached files.

Reviewer #1: **Yes: **Isamu Okada

Reviewer #2: No

Reviewer #3: No
---

## [Decision Letter · Decision Letter 1]

28 Feb 2022

Dear Dr. Salahshour,

Thank you very much for submitting your manuscript "Interaction between games give rise to the evolution of moral norms of cooperation" for consideration at PLOS Computational Biology. As with all papers reviewed by the journal, your manuscript was reviewed by members of the editorial board and by several independent reviewers. The reviewers appreciated the attention to an important topic. Based on the reviews, we are likely to accept this manuscript for publication, providing that you modify the manuscript according to the review recommendations.

The paper was once again reviewed by the original reviewers. The reviewers acknowledge that extensive revisions were made to address their previous concerns, appreciate the effort by the author, and are more positive about this manuscript.

One important issue still raised, however, has to do with the style and clarity of presentation. Reviewer #2 provides ingenious and very constructive suggestions on how to address that, while agreeing that the paper now offers a better intuition for what drives the observed results. Given the positive appreciation of Reviewer #1 and Reviewer #3, we recommend that the paper is accepted conditionally on minor revisions to address the pertinent presentation issues raised by Reviewer #2. Reviewer #1 provides a pointer to a recently published paper on the evolution of norms and moral preferences, which seems related with the submitted manuscript.

Sincerely,

Fernando Santos, Ph.D.

Guest Editor

PLOS Computational Biology

Ville Mustonen

Deputy Editor

PLOS Computational Biology

[LINK]

Thank you for the comprehensive revisions made to address the issues raised by the reviewers. The paper was reviewed by the original reviewers. The reviewers acknowledge that extensive revisions were made to address their previous concerns, appreciate the effort by the author, and are more positive about this manuscript.

One important issue still raised, however, has to do with the style and clarity of presentation. Reviewer #2 provides ingenious and very constructive suggestions on how to address that, while agreeing that the paper now offers a better intuition for what drives the observed results. Given the positive appreciation of Reviewer #1 and Reviewer #3, I recommend that the paper is accepted conditionally on minor revisions to address the pertinent presentation issues raised by Reviewer #2. Reviewer #1 provides a pointer to a recently published paper on the evolution of norms and moral preferences, which seems related with the submitted manuscript.

Reviewer's Responses to Questions

**Comments to the Authors:**

Reviewer #1: I have confirmed that the revised manuscript has been significantly improved in response to all comments including myself, and thus I recommend the publication.

Reviewer #2: GENERAL EVALUATION.

The paper explores what happens if individuals engage in two consecutive games. The first game is a social dilemma (prisoner's dilemma); the second game is an anti-coordination or coordination game (e.g., a snowdrift game or a stag-hunt game). The author shows that this coupling of games can have two benefits:

(1) It can increase cooperation in the first game

(2) It can avoid coordination failures in the second game

Already in my first report, I argued that the general setup is interesting and worth studying. Also, I argued that the author has put extensive work into his analysis, by exploring various different scenarios (e.g., varying the exact anti-coordination game to be played as the second game; exploring both direct and indirect interaction scenarios; comparing well-mixed and lattice-based populations; replicator dynamics and simulations, etc).

On the downside, I argued that it is sometimes hard to understand the results, because they often feel like coming out of a black box. To address these weaknesses, I suggested various clarifications. For example, I suggested that the author offers an explicit (Nash) equilibrium analysis. Also, I suggested that the author also considers the case in which the second game is a coordination game, rather than an anti-coordination game. The author has taken all these suggestions into account. I believe the intuition for what drives the observed effects is now clearer, and I would like to thank the author for his efforts.

In some sense, I believe the manuscript contains actually too much information for a typical journal article. By considering so many distinct scenarios, it is somewhat difficult for the reader to fully grasp the nature of all results.

I believe it would have been useful to write a first paper only about the baseline case, direct interactions in well-mixed populations. This case could then have been analysed in full detail, and the results could have been explained more clearly.

Having said that, I believe also the current article is fine. In the following, I only make a few minor suggestions to increase the clarity of the text.

COMMENTS:

[Line and page number references refer to the version with annotated changes]

(1) In the abstract, the author claims that "coupling between the two games emerges and helps to solve the dilemma"

I would prefer if the author were more careful here. Coupling does not fully solve the dilemma (as it typically does not lead to full cooperation in the prisoner's dilemma, but only say 60%). Please describe more clearly that coupling may promote cooperation; however, at least if the second game is an anti-coordination game, full cooperation in the first game still requires additional mechanisms. Moreover, more cooperation in the first game might make it more difficult to anti-coordinate in the second game.

(2) Similarly, in the introduction it would be useful if the author more clearly explains what a "cooperative equilibrium" is. These equilibria are typically not fully cooperative; they only exhibit *some* cooperation in equilibrium.

(3) At the end, of the introduction, the author states: "In contrast, when game B is a coordination game, moral norm of cooperation does not evolve in a structured population."

I would prefer if the author chooses a more humble formulation, such as: "In contrast, in those simulations in which game B is a coordination game, we did not observe the evolution of cooperation."

The reason why I suggest this is the following: the author's claim might be true for the simulations he performed. However, I could imagine several small modifications of the model such that cooperation might actually evolve even if the second game is a coordination game. For example, the author could have allowed for rare implementation errors in the first game, such that cooperators defect with some small probability epsilon. In that case, Cdu becomes a strict Nash equilibrium. Hence Cdu might also evolve in structured populations.

(4) On page 5, line 150, I wouldn't motivate 1-eta as the likelihood of players "guessing" their co-player's previous action. Rather, I would say it is the likelihood that players are correctly informed about their co-player's previous action.

(5) It would be good if the author explains more clearly early on which default payoff values are used for the different games. For example, in Table 1, it would be useful to see the base game payoffs at least in the figure caption.

(6) In Figure 2b, the author suggests that cooperation can emerge if the cost of cooperation is smaller than the "cost of coordination failure". However, game theorists usually do not interpret R_B - P_P as the cost of coordination failure in a stag-hunt game (rather coordination failure would be typically associated with the risk of getting one of the non-equilibrium payoffs T_B or S_B). Rather R_B - P_B is the cost of coordinating on the payoff-dominated equilibrium.

(7) In Figure 3, are these all possible fixed points, or is it just a selection? Also, it would be useful if the author always speaks of the replicator-mutator dynamics (as standard replicator dynamics wouldn't allow for mutations). In addition, it would be useful if the author points out that he uses a discrete-time replicator dynamics (because I'd assume most readers would associate replicator dynamics with the continuous version).

(8) On page 8, the author uses formulations like fixed point "SH1". Please define SH1 at its first occurrence.

(9) The caption of Figure 5 does not match the figure (panel a seems to have switched places with panels b and c).

(10) Line 417; please correct "p_C^initial.

(11) When using an initial condition in which all strategies are played with equal frequency, please do not speak of a "random initial population". This initial population is not randomly chosen, it is a very specific initial population. Instead one may speak of the "center" initial population.

(12) On page 18, I would be careful attributing the differences between simulations and replicator dynamics only to "finite size effects". After all, the simulations differ from replicator dynamics in more than just the finite size of the population. It is not clear that it is the finite size of the population that is responsible for the differences.

(13) Please do not speak of "equilibrium fixed points" and "non-equilibrium fixed points". After all, both types of fixed points correspond to Nash equilibria. It is somewhat strange to call the second Nash equilibrium a "non-equilibrium fixed point". Please find a different name to refer to the two fixed points.

(14) In general, it would be helpful if the axis labels of the figures do not only mention the variable names (like "eta" or "rho_C"), but also some text that describes this variable (like "Error rate" or "Probability to cooperate in first game"). It is hard for a reader to keep all variables in memory, so any help here is appreciated.

(15) Please avoid referring to second game behavior as "cooperation" (as on page 23, line 739). The text becomes clearer if only first game behavior is interpreted as cooperation.

(16) Page 25: There are multiple instances of "see of" which should read "sea of"

(17) In the discussion, I would prefer if the author rephrases the sentence "This appears to provide a possible explanation for the evolution of morality"

Instead of "explanation", I would prefer "a possible mechanism". In my opinion, this would make more clear that the proposed model certainly does not explain all aspects of morality.

(18) In the discussion, what exactly does the author mean when saying that "assessment module and action module need to be placed on the same level"? Please rephrase.

Also, please note that there are indirect reciprocity models where players only need first-order information (for example the models of Ulrich Berger, or the model of the recent PNAS paper of Drew Fudenberg and colleagues, on "simple records").

(19) As a general piece of advice, I think an article like this would be easier to read if the author engaged more in meta-communication with the reader. Like, in the beginning of a paragraph, make sure the reader understands what the paragraph will be about. At the end of the paragraph, make sure the reader understands what the take-home message is.

Reviewer #3: I plause the author for a very competent revision, which addressed all my comments. I think that the paper can be published. Just one minor comment, that the author may or may not follow: a review article on the role of moral norms on cooperation and other forms of pro-sociality has been recently published, I think that the author may find it interesting for his research:

https://royalsocietypublishing.org/doi/full/10.1098/rsif.2020.0880

**Have the authors made all data and (if applicable) computational code underlying the findings in their manuscript fully available?**

Reviewer #1: Yes

Reviewer #2: Yes

Reviewer #3: None

PLOS authors have the option to publish the peer review history of their article (what does this mean?). If published, this will include your full peer review and any attached files.

Reviewer #1: **Yes: **ISAMU OKADA

Reviewer #2: No

Reviewer #3: No

Figure Files:

Data Requirements:

Reproducibility:

References:

---

## [Editor Report · Decision Letter 2]

21 Jul 2022

Dear Dr. Salahshour,

We are pleased to inform you that your manuscript 'Interaction between games give rise to the evolution of moral norms of cooperation' has been provisionally accepted for publication in PLOS Computational Biology.

Best regards,

Fernando Santos, Ph.D.

Guest Editor

PLOS Computational Biology

Ville Mustonen

Deputy Editor

PLOS Computational Biology

I believe that the author has addressed all (mainly presentation) points raised by the reviewers and this manuscript can now be accepted for publication in PLOS Computational Biology.

---

## [Editor Report · Acceptance letter]

25 Aug 2022

PCOMPBIOL-D-21-00570R2 

Interaction between games give rise to the evolution of moral norms of cooperation

Dear Dr Salahshour,

I am pleased to inform you that your manuscript has been formally accepted for publication in PLOS Computational Biology. Your manuscript is now with our production department and you will be notified of the publication date in due course.

With kind regards,

Zsofia Freund
